# Explainable chemical artificial intelligence from accurate machine learning of real-space chemical descriptors

Miguel Gallegos [1], Valentin Vassilev-Galindo [2], Igor Poltavsky[3], Ángel Martín Pendás [1] ✉ & Alexandre Tkatchenko [3] ✉

Machine-learned computational chemistry has led to a paradoxical situation in which molecular properties can be accurately predicted, but they are difficult to interpret. Explainable AI (XAI) tools can be used to analyze complex models, but they are highly dependent on the AI technique and the origin of the reference data. Alternatively, interpretable real-space tools can be employed directly, but they are often expensive to compute. To address this dilemma between explainability and accuracy, we developed SchNet4AIM, a SchNet-based architecture capable of dealing with local one-body (atomic) and two-body (interatomic) descriptors. The performance of SchNet4AIM is tested by predicting a wide collection of real-space quantities ranging from atomic charges and delocalization indices to pairwise interaction energies. The accuracy and speed of SchNet4AIM breaks the bottleneck that has prevented the use of real-space chemical descriptors in complex systems. We show that the group delocalization indices, arising from our physically rigorous atomistic predictions, provide reliable indicators of supramolecular binding events, thus contributing to the development of Explainable Chemical Artificial Intelligence (XCAI) models.

Chemistry, similar to many other scientific disciplines, is now inextricably linked to computer-assisted simulations. For instance, electronic-structure calculations have become so deeply rooted in chemistry that they are used today as a standard characterization technique in the chemist's toolbox, on par with NMR spectra or X-ray diffraction data. The applications of electronic-structure theory, together with its coupling to molecular dynamics[1], have revolutionized materials science[2], catalysis[3], photochemistry[4], and chemical reactivity[5], to name just a few examples. Many of the pressing efforts today have been devoted to increasing the efficiency and accuracy of electronic-structure methods that deliver reliable molecular quantities, such as energetic or response properties. However, this has resulted in an unintended consequence: it is now far easier to get accurate property predictions than to interpret them physically or chemically. In this sense, and as some of us have recently put forward[6], Coulson's maxim *"give us insight not numbers"*[7] has been set aside. Today, most researchers use techniques in their essential post-calculation interpretation steps that are based on much cruder assumptions than those used in the simulations themselves. This undesirable situation can be ameliorated by resorting to orbital invariant chemical descriptors, which provide physically meaningful properties and can be obtained either in real or in momentum space.

These descriptors are based on condensing the information content of the wavefunction through reduced density matrices of various orders evaluated at a set of chemically relevant points of space, typically through the analysis of the topology induced in real space by their gradient fields, in what it is usually known as quantum chemical topology[8]. An important ingredient of the latter is isolating atomic (or

[1]Department of Analytical and Physical Chemistry, University of Oviedo, E-33006 Oviedo, Spain. [2]IMDEA Materials Institute, C/Eric Kandel 2, 28906 Getafe, Madrid, Spain. [3]Department of Physics and Materials Science, University of Luxembourg, L-1511 Luxembourg City, Luxembourg. ✉e-mail: ampendas@uniovi.es; alexandre.tkatchenko@uni.lu

functional group) regions from wavefunctions. Indeed, such a topic has sparked a lot of interest in the scientific community in recent years which has crystallized in a plethora of different decomposition and partitioning techniques grounded on countless theoretical frameworks. Notoriously relevant examples of these include symmetry-adapted perturbation theory[9] or even the celebrated density-functional theory[10,11], to name just a few. In this regard, a particularly rigorous way to achieve this partitioning comes from the so-called Quantum Theory of Atoms in Molecules (QTAIM) of R. F. W. Bader[12]. QTAIM metrics allow for an unbiased account of physical and chemical processes at any level of theory, regardless of the use of atom-centered, plane-wave, or numerical basis sets. On one side, they provide access to electron counting properties, including electron populations and their statistics[13], the key to a proper understanding of the role of electron localization and delocalization, for instance. On the other, thanks to the underlying QTAIM partitioning of the physical space, every observable is also additively reconstructed from domain contributions.

For instance, and if we restrict ourselves to standard Coulomb Hamiltonians, the topological decomposition of the energy, the so-called interacting quantum atoms (IQA) approach[14], provides an exact partition in terms of intra- and interatomic components. Contrary to other modern energy decomposition analyses, like those tracing back to Morokuma's seminal insights[15,16], IQA does not depend on artificial intermediate states nor on single-determinant approximations to the wave function. In contrast to other highly accurate procedures, like symmetry-adapted perturbation theory[9], it can also be applied to arbitrary molecular geometries, both in weakly and strongly interacting regimes. Since QTAIM and IQA provide physically meaningful properties, having been used to shed light on an ever-growing set of problems (see, e.g., refs. [17–19]), many have envisioned their usefulness in the construction of better-rooted, general-purpose tools, including force-fields. Unfortunately, this rigor comes at the expense of a considerable computational cost. Since the resultant atomic regions have no analytical definition, the evaluation of domain expectation values of two-electron contributions implies the calculation of a vast number of expensive 6D numerical integrations[20].

Although crucial work is still being currently done in the development of faster algorithms[21], a possible answer to this problem lies in the exploitation of machine learning (ML). The extraordinary prediction abilities of artificial intelligence (AI) approaches such as Deep Learning, or ML in general, are reshaping the course of modern research. In fact, the large efficiency of state-of-the-art AI brings the possibility of accurately performing complex tasks in feasible time scales. The success of this field is directly evidenced by the outburst of AI tools in the modeling of virtually any physico-chemical property[22–25] ranging from molecular structure[26–28], energy landscapes[29,30], spectroscopic transitions[31], aromaticity and reactivity trends[32], magnetism[33], mechanical features[34], or even chemical fragrances[35], to name just a few examples. As such, the implementation of AI marks a crucial step forward in many realms, such as the fields of drug discovery[36] and materials design[37], showcasing its ability to lead modern scientific and technological research. As far as Quantum Chemical Topology is regarded, much effort has been devoted in the last two decades to implement ML in the QTAIM or IQA domains. In this context, it is worth highlighting the work of P. Popelier and coworkers[38–42], who used Gaussian Process Regression to ease the computation of multiple terms including atomic energies, charges or multipole moments. Albeit kernel models have been the method of choice in this field, neural networks (NNs) have also been employed. For instance, P. Popelier developed an NN model for the accurate estimation of atomic multipole moments of water clusters[43]. Similarly, some of us have recently reported a simple atomistic NN for the prediction of QTAIM partial charges of gas-phase main-element (C, H, O, and N) compounds[44].

However, the growing intricacy of the problems tackled by ML, together with the scarcity of uncertainty metrics[45], makes it prone to suffer from the Clever Hans effect: getting the right answer from the spurious interplay of wrong reasons. As such, illuminating the obscure nature of AI has become of paramount importance, as reflected by the so-called Physics-guided or inspired models[46] or the emerging field of explainable artificial intelligence (XAI)[47,48]. In spite of lacking an exact definition[49], XAI imagines AI as a tool to unravel the nature of the problem under study and its underlying rationale, exposing the actual capabilities and limitations of the models. Although a growing interest has emerged in recent years[45,47], the application of XAI to chemistry is still in its infancy owing, among other things, to the far from trivial encoding of the physical variables of a system into a machine-interpretable array, the so-called chemical featurization problem. Often, a fixed functional form is used to compute these chemical features, a strategy that requires fine-tuning a considerable number of hyperparameters. Contrarily to these hand-crafted descriptors, end-to-end models learn the atomistic representations directly from more tangible physical variables (e.g., $R, Z$). One of these is SchNet[50–52], which has gained wide popularity due to its versatility in accurately predicting different quantum chemical properties. The flexible representation scheme provided by SchNet, achieved through continuous filtering generators along with the state-of-the-art performance of Deep NNs, provides extraordinary inferring abilities in countless applications[53–58]. However, the default SchNet architecture was designed to predict global quantities. Instead, dealing with local chemical properties requires adjustments to the SchNet model.

In this work, we introduce an essential modification of this approach, named SchNet4AIM and implemented in the SchNetPack (SPK) package[52], targeting predictions of local quantities, including both atomic (one-body or 1P in what follows) and pairwise (two-body or 2P) terms. As such, this constitutes a step forward in the development of end-to-end ML chemical tools in the context of local quantum chemical analysis. The performance of the resultant architecture is put to the test for a set of QTAIM and IQA descriptors such as atomic charges ($Q$), localization ($\lambda$), and delocalization ($\delta$) indices as well as IQA energetic terms. The relevance of these terms, some of which have been proven reliable estimators of a wide range of molecular properties (e.g., aromaticity, $pk_a$, or spectroscopic features, to name a few)[59–61], extends well beyond the QCT realm. The strategy largely outperforms previously reported ML@QTAIM general-purpose approaches[44]. Besides higher accuracy, SchNet4AIM affords robust models while requiring much less training data than in previous attempts. Moreover, we show how the rigor of real space techniques grants the models with promising extrapolation and transferable capabilities, exhibiting physically coherent behaviors that enable the distillation of valuable chemical insights way beyond the mere quantitative values. In this way, the rigor of the underlying physical models and the flexibility of SchNet4AIM puts our approach in a privileged spot in the obtention of physically-behaved and understandable outputs. Unlike the usual scenario, where chemical properties obtained by ML are derived from the aggregation of meaningless, black-box-derived quantities, SchNet4AIM's molecular predictions can be traced back to physically rigorous atomic or pairwise terms, enabling the distillation of valuable insights and interpretations. Thereby, and following Coulson's maxim, we introduce a clear example of an explainable chemical AI model (XCAI). More importantly, this is done without requiring the use of extrinsic explainability tools, but it is instead inherent to the combination of SchNet4AIM with rigorous local quantum chemical properties. As a proof of concept, we show how this approach breaks the QTAIM/IQA computational bottleneck by allowing a general user to follow the evolution of otherwise prohibitively expensive quantum chemical descriptors along relevant chemical processes. This is done on the fly, at negligible extra computational cost, and opens new opportunities to rigorously interpret the result of

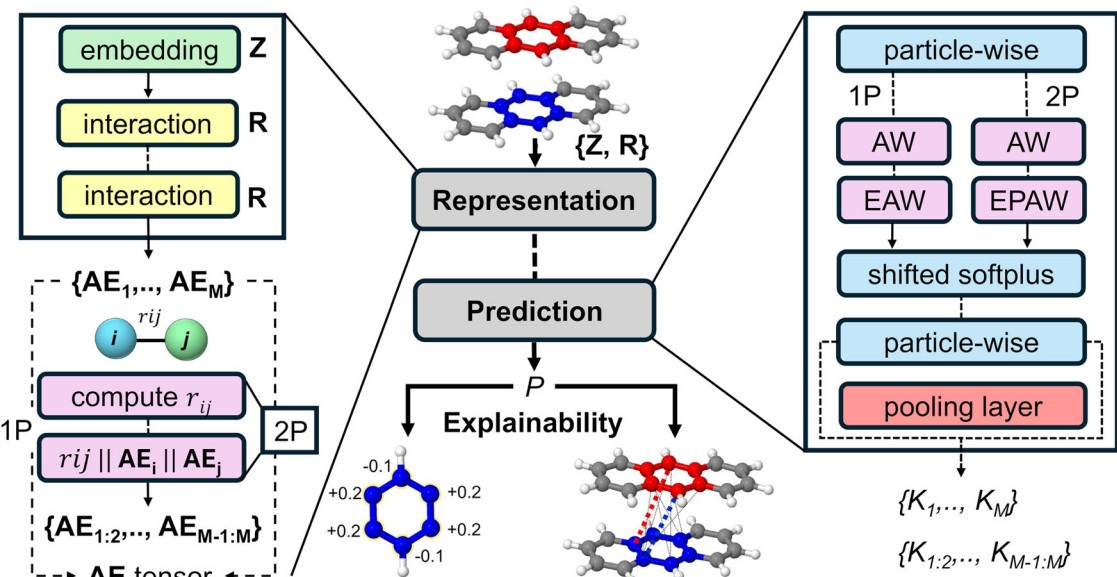

**Fig. 1 | Architecture of SchNet4AIM (modified SchNetPack toolbox).** Schematics of the SchNetPack toolbox targeted for the prediction of one-body (1P) and two-body (2P) terms, showing the main representation (left) and prediction (right) blocks. The atomic environment (**AE**) tensor, created by the representation block, describes an atom or atomic pair in a molecule. This is parsed to the prediction block which outputs a molecular property (*P*) decomposed into a collection of locally defined terms (*K*). The labels **Z** and **R** denote atomic numbers and positions of an *M* atom system, while $r_{ij}$ signifies the distance between atoms *i* and *j*. AW, EAW and EPAW refer to AIMwise, ElementalAIMwise, and ElementalPairAIMwise output modules, respectively, used to construct universal and particle-specific models.

large-scale chemical simulations. To this end, we demonstrate that electron delocalization metrics are reliable indicators of supramolecular binding events, offering a detailed picture of the electron rearrangements driving these intricate complexation phenomena.

## Results

### Algorithmic details and architecture

The current section gathers the modifications made to the SchNetPack toolbox to enable the prediction of local quantum chemical descriptors using SchNet4AIM. Details on the original SchNet architecture and SchNetPack software can be found in Supplementary Note 1 or in the literature[50–52]. A schematic representation of the architecture of our modified version of the toolbox is shown in Fig. 1.

Just as in the original version, the representation block maps the molecular information (geometries and atomic numbers, *R* and *Z*) into a SchNet-like descriptor of fixed size (*n*), resulting in *M* local atomic environments, {**AE**$_1$,...,**AE**$_M$}. In the case of 1P properties, this can be readily used to feed the atomistic NN models, whereas the latter must be further transformed to obtain a pair-wise descriptor. Although it is possible to envision different approaches to describe a given atomic pair, we have opted for the concatenation of atom-wise features along with the pairwise distance to build the 2P vectors. The inclusion of the latter has been motivated by the strong dependence of many two-body properties on the interatomic distance (Supplementary Fig. 4). Furthermore, the resultant features exhibit the desirable permutational, translational, and rotational invariances. All of this is handled by the SPK.atomistic.model which, in the case of 2P properties, obtains the atomistic vectors (**AE**$_i$, **AE**$_j$) and the interatomic distances, $\|\mathbf{r}_i - \mathbf{r}_j\|$ to reconstruct the final pair-wise featurization descriptor, {**AE**$_{1:2}$,...,**AE**$_{M-1:M}$}, containing a total of $M(M-1)/2$ non-equivalent elements (excluding the diagonal contributions). We note in passing that each of the pair-wise vectors, **AE**$_{i:j}$, has now a length of $2n+1$ elements. Thus, moderate values of *n* should be used to prevent excessively large input vectors which can result in complex and poor performing models.

The resultant particle-wise descriptor, here referred to as the AE tensor, is then fed into the prediction block to train the NN models.

Three different local output models have been implemented in the SPK.atomistic module by removing the cumulative pooling layer commonly used in the prediction of molecular observables: AIMwise, ElementalAIMwise, and ElementalPairAIMwise. The former, which can be either used for 1P or 2P quantities, employs a single NN model for all the particles. On the other hand, the ElementalAIMwise (1P) and ElementalPairAIMwise (2P), create a collection of *K* particle-type specific models. *K* is given by either the number of non-equivalent chemical atom types (*T*) or their pairwise combination $T(T+1)/2$ for 1P and 2P properties, respectively. Building 2P-specific models raises the need to create unique, yet permutationally invariant, identifiers for each chemically different pair. To solve this issue, we use a symmetric matrix, pairmat, of non-repetitive integers obtained from the $Z_i$ and $Z_j$ atomic numbers of each pair (further details can be found in Supplementary Note 2). The remaining architecture involved in the prediction was essentially left unmodified, and thus, it will not be discussed. Further details about the modifications made to the SchNetPack toolbox and its features can be found in Supplementary Note 2. Finally, we also note in passing that it is possible to exploit the physics governing the local quantum chemical properties learned by SchNet4AIM to estimate the reliability of its predictions. As such, one can devise different uncertainty estimates, as detailed in Supplementary Note 2. For instance, in the particular case of the QTAIM electron metrics used in this work, the error in the reconstruction of the molecular electron count (*N*) provides a very convenient way of doing so.

### Initial performance tests

Let us start by assessing the performance of SchNet4AIM when trained on local properties using the IQA energies of a water cluster database, details on the latter can be found in Methods. For the sake of simplicity, we will focus on the more diverse and robust energetic properties of O atoms, and in particular, the kinetic energy ($T^O$) and the pairwise interaction energy with neighboring, intra-, and inter-molecular, H atoms ($E^{O-H}_{inter}$). Figure 2 gathers their dispersion plots, as predicted by SchNet4AIM, using universal (AIMwise) and particle-specific (ElementalAIMwise or ElementalPairAIMwise) models. From the latter, it is

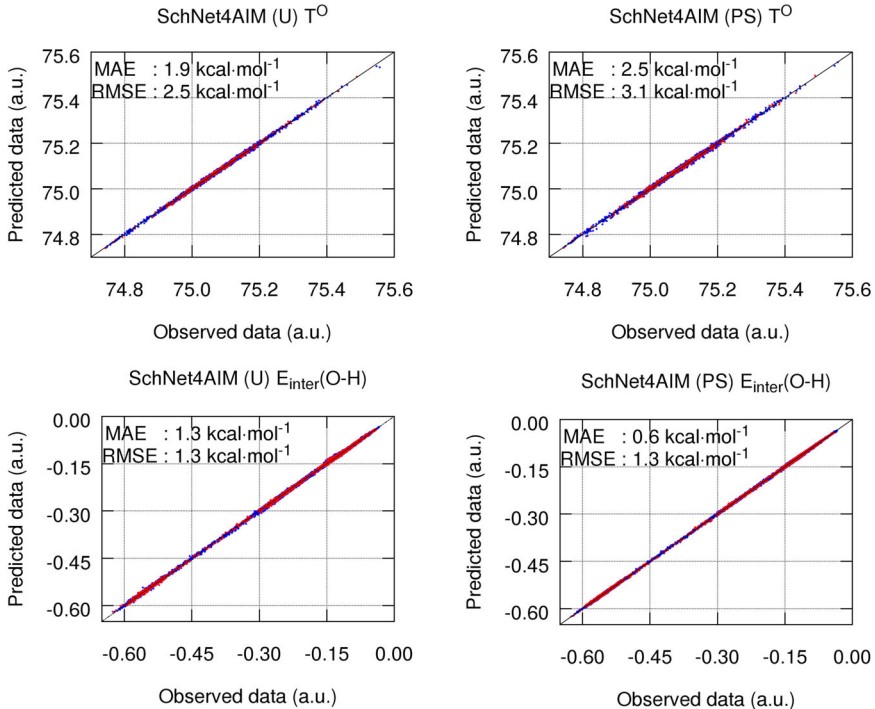

**Fig. 2 | Dispersion plots for $T^O$ and $E_{inter}^{O-H}$.** Dispersion plots for the kinetic energy of the O atoms, $T^O$, and O–H interaction energies, $E_{inter}^{O-H}$, of the water cluster database. The training and testing data points are shown in blue and red, respectively. All values are reported in atomic units (a.u.), whereas the testing error metrics are given in kcal mol⁻¹. The mean absolute error (MAE) and root mean squared error (RMSE) were employed as measures of accuracy. The labels U and PS are used to refer to universal (AIMwise) and particle-specific (ElementalAIMwise or ElementalPairAIMwise) models, respectively. Source data are provided as a Source Data file.

evident that SchNet4AIM is able to accurately predict atomic quantum chemical properties, exhibiting very reasonable MAEs of just under 2.0 kcal mol⁻¹ (≈0.003 a.u.) for the kinetic energy of the O atoms (with values spanning over a range of roughly 0.8 a.u.).

Moving now on to the slightly more intriguing scenario of interatomic properties, and using the $E_{inter}^{O-H}$ term as a prototypical example, reveals quite similar findings pushing the errors down to the noise bound (≤1.0 kcal mol⁻¹) with MAEs as low as 0.3 kcal mol⁻¹ in the particular case of intra- and inter-molecular H–H interactions (Supplementary Tables 8 and 9). As far as the output model is regarded, particle-specific architectures perform slightly better than universal models, something which is accentuated for certain 2P terms (e.g., $E_{inter}^{H-H}$, Supplementary Figs. 2 and 3). Particle-specific models are expected to outperform single NN approaches as the chemical complexity of the system is increased since a large number of particle types may selectively bias the model's performance toward certain atoms or atomic pairs and thus decrease the net prediction accuracy. Altogether, these results evidence the suitability of SchNet4AIM for the accurate computation of local properties.

## SchNet4AIM in extrapolation domains

After showing the ability of SchNet4AIM to compute 1P and 2P properties, we will now evaluate its extrapolation abilities. Generally speaking, ML models can only be used within those regions of the chemical space that have been sampled during their training, the so-called interpolation domains. Instead, their predictions often become erratic as one moves away from the latter, delving into the extrapolation region. Although this limitation is intrinsic to the field, dealing with quasi-transferable properties can ameliorate such a limitation. In this regard, it is worth mentioning that our approach is particularly fitting to this task owing to the transferable nature of the QTAIM attraction basins and their local properties[62–64]. In this context, chemical transferability has actually been evaluated in the past[65,66], with numerous studies devoted to exploring the extent of the quasi-transferable nature of a wide collection of topological[63,67–70] and geometrical features[71]. This is particularly interesting as, if successful, it could extend the applicability of the SchNet4AIM models to much more complex scenarios than those used throughout the training.

To explore this idea, we will focus on the more diverse electronic QTAIM properties of CHON molecules: the local electron counts ($Q$) along with their localized ($\lambda$) and delocalized ($\delta$) contributions. SchNet4AIM was trained on a subset of the CHON database, see Methods for further details, comprising ground-state minima of the potential energy landscape. Then, its performance was explored by studying a chemical reaction involving very far from equilibrium structures, and comprising thus a clear case of extrapolation. In interpolation domains, our model outperforms previously reported approaches[44] when predicting the atomic charges ($Q$) while using only 10% of the training data required by the latter (Supplementary Note 11). Such a result is not limited to the local values (with MAE errors in the mili-electron range), but SchNet4AIM is also able to recover the electro-neutral character of the molecules with much higher accuracy ($10^{-3}$ to $10^{-2}$ e⁻), evidencing the physical-behavior of its local components (Supplementary Table 19). Furthermore, the prediction accuracy of SchNet4AIM drops slowly with the size of the system, even when going beyond the largest molecular size explored during the training (Supplementary Fig. 14). Altogether, the aforementioned observations, also found for the localization ($\lambda$) and delocalization indices ($\delta$), suggest that SchNet4AIM affords more accurate local properties with the ability to better reconstruct the molecular quantities.

Since SchNet4AIM was only trained on the near-equilibrium CHON space, we have decided to employ a chemical reaction, see Fig. 3, gathered from the literature[44] as a prototypical extrapolation scenario: such a transformation involves out-of-equilibrium structures, very far from the potential wells sampled during the training. The initial stages of the reaction involve barely any electron fluctuations, as

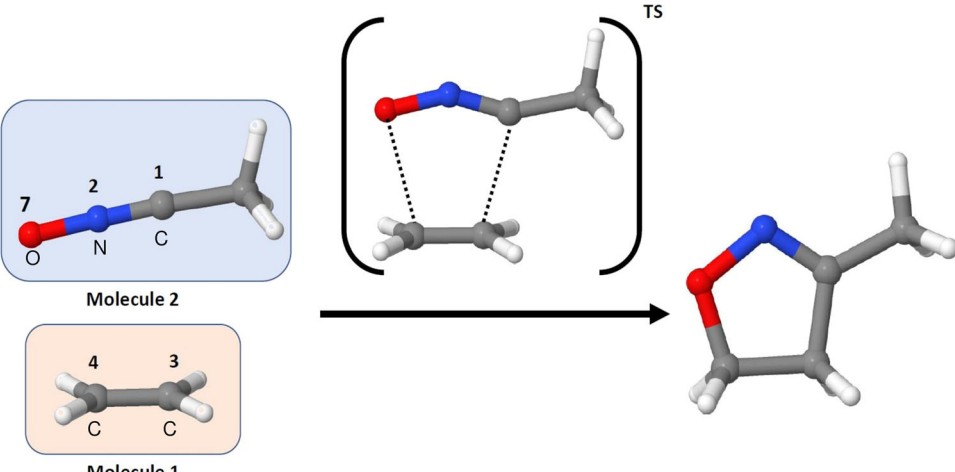

**Fig. 3 | 1,3-dipolar cycloaddition.** Schematic representation of the addition of acetonitrile oxide to ethylene to yield a 5-membered heterocycle, showing the transition state (TS) structure involved in the transformation. The numbers and labels of the main atoms involved in the reaction are shown. The remaining, and with the exception of the C atom in the $CH_3$ moiety of acetonitrile oxide, are H atoms.

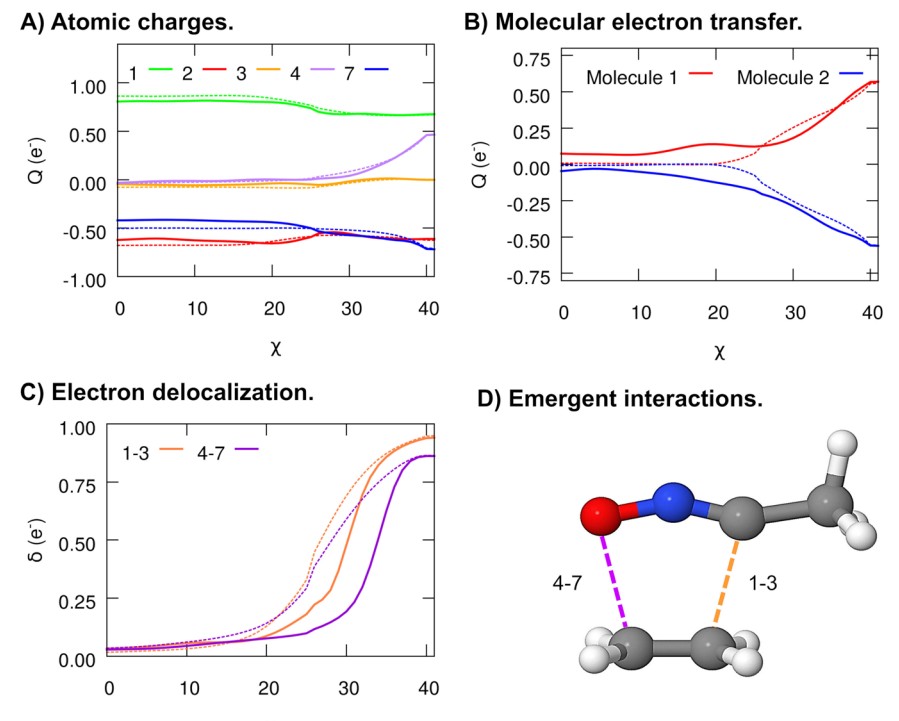

**Fig. 4 | SchNet4AIM predictions in extrapolation regimes: electronics of a 1,3-dipolar cycloaddition.** Evolution of the atomic charges of the main atoms involved in the reaction (**a**), the total molecular charge of each of the reactants (**b**), and the electron delocalization between the terminals atoms (1,3 and 4,7) of both species (**c**). Atomic or molecular charges are denoted as $Q$, whereas the term $\delta$ is used to refer to the delocalized electron counts. Solid and dashed lines are used to represent the predicted (SchNet4AIM) and observed data, respectively. The SchNet4AIM models are tested in never-seen (extrapolation) regions of the chemical space. All values are reported in electrons (e⁻) relative to the progress of the reaction coordinate ($\chi$). The atomic labels correspond to the numbering shown in Fig. 3. **d** The transition state of the reaction, showing the emergent C–C and C–O bonds that will be formed, is also shown. Source data are provided as a Source Data file.

evidenced by the fairly monotonous behavior in the atomic charges of the main atoms involved, comprised in Fig. 4A. It is only close to the transition state ($\chi = 25$) where major changes in $Q$ start to be observed: atom 1 (C) experiences a moderate electron enrichment, also found in a more subtle way for atom 7 (O), at the expense of depleting atom 4 (C). The interplay between the fluctuating atomic charges accompanying the formation of the final product promotes a net transfer of ≈0.6 electrons from ethylene to the acetonitrile oxide, as shown in

Fig. 4B. These findings, being in accordance with the quantum-mechanical calculations, arise as a result of the large electronegativity difference between the interacting fragments. These observations are noteworthy on their own, as they evidence the remarkable generalization abilities of our approach, offering accurate predictions even in extrapolation regimes.

Naturally, the aforementioned changes are accompanied by prominent shifts in the delocalized electron counts: the $\delta(1, 3)$ and $\delta(4, 7)$

metrics peak at about 0.95 and 0.85 electrons, respectively, evidencing the formation of the C–C and C–O single bonds (see Fig. 4D) of the final product, as shown in the Fig. 4C. Taking a closer look at the qualitative trends reveals that the 1–3 bond is formed prior to the slightly more latent 4–7 interaction, proving that SchNet4AIM is able to predict the subtle lack of synchronicity predicted by quantum chemical calculations. Altogether, our model captures, at a qualitative and quantitative level, most of the chemical insights afforded by the more expensive quantum chemical calculations even outside the interpolation regimes used for its training. These findings show that the rigor of QCT and the reliability of the underlying SchNet4AIM architectures yield transferable ML@QTAIM models with decent extrapolation abilities. Besides accurate, SchNet4AIM predictions entail a negligible computational cost when compared to conventional (quantum chemical) approaches, leading to speedup factors of up to $10^4$ for a 48 atoms system, see Supplementary Note 17 for further details. Besides this, we also note in passing that SchNet4AIM shows quickly-converging learning curves, as gathered in SI Supplementary Note 19. Such a finding evidences its ability to adequately generalize the patterns learned from even a small set of molecular instances to unseen data.

It should also be noted that the extrapolation capability and transferability of SchNet4AIM are subject to three main limiting factors. First, the size and quality (e.g., how well the chemical space has been sampled) of the reference data. Second, the representation hyperparameters: opting for a smaller cutoff radius may improve transferability at the cost of reduced accuracy in describing the local environments of atoms or atomic pairs. Finally, the extrapolation capability is closely linked to the complexity of the target properties and the impact that the local environment has on the latter. In this way, more chemically complex systems are more difficult to predict, as evidenced by the subtle decrease in prediction accuracy found when extrapolating. For instance, the MAEs in the estimation of the atomic charges increase to the $10^{-2}$ electrons range in the case of chemical reactions (see Supplementary Note 13). In these particularly complex scenarios, using more local and compact SchNet4AIM representations, as well as increasing the global-to-local tradeoff in the loss function, can improve the generalization abilities of the models, ameliorating these inconveniences. Further details are given in Supplementary Note 13.

## Chemical insights from SchNet4AIM predictions

Finally, and given the promising performance shown by SchNet4AIM in the computation of local quantities, we will now proceed to explore how this model can be used to distill valuable chemical insights. We will focus on interpretability, showing how the synergy between SchNet4AIM and QTAIM can provide physically coherent outputs in line with XCAI models in way more complex scenarios than those used in the training of the underlying models. For such a purpose, the SchNet4AIM models trained on the QTAIM metrics of small size CHON molecules in their equilibrium configurations (detailed previously), will be used to disentangle the intricacies behind a complex supramolecular process. This constitutes an ideal scenario to test the generalization abilities and transferability of SchNet4AIM while showing how the latter can shed light on intricate chemical phenomena through its intrinsically explainable outputs. As a proof of concept, this will be exemplified with the $CO_2$ capture and release by a recently reported Calix[4]arene[72], referred to as 13P, from now on. The latter is equipped with $NH_2$ groups that enable the formation of a hydrogen bond (HB) driven supramolecular cage, within which the $CO_2$ can get easily trapped and subsequently released.

In recent years, cyclic oligomers, such as Calix[n]arenes, have proven to be valuable receptors for a wide variety of ligands[73–76]. Their high affinity and modular backbone have motivated the development of tailor-made macrocycles for numerous applications[77–81], including gas sensing and capturing devices[82–84]. Indeed, and since it was first

reported in 1991[85], the complexation and fixation of $CO_2$ by aromatic macrocycles has become one of the most promising strategies behind emerging $CO_2$ capture technologies[86–91]. Finding the best-performing receptors is a far from trivial task, which can benefit from modern computational tools with the ability to dissect the chemical interactions governing the complexation phenomenon. Although some trends can be inferred from geometrical features, a more in-depth picture often requires expensive tools, such as electronic metrics (e.g., $\delta$), whose prohibitive computational cost prevents their use in dynamic scenarios.

In this section, we will show how the SchNet4AIM predicted electron delocalization offers a robust analysis of the driving forces governing the $CO_2$-13P binding, paying special attention to how the interpretable SchNet4AIM outputs can monitor this cage-opening and $CO_2$ release phenomena while showing the specific ligand–receptor interactions that drive the binding events. At this point, it should be stressed that this is a particularly challenging test since SchNet4AIM, trained on isolated and small-size molecules in static configurations, will be forced to make predictions on a dynamic scenario involving a large-size and non-covalently bonded system. In this way, not only will our model be extrapolating, but it will also be acting on completely different chemical spaces and energy landscapes than those seen throughout the training, evidencing the transferability of the latter. All the insights shown in the upcoming discussions are based solely on the electron delocalization between relevant functional groups and chemical moieties obtained from the aggregation of raw $\delta$ SchNet4AIM outputs. The local build-up of delocalized electron population between two groups, corresponding to a maximum in the $\delta$ values, will be used as an indication of a binding event. Furthermore, we will show how SchNet4AIM can easily identify the most relevant pairwise terms that dominate these events, evidencing the intrinsic explainability of our approach. In this way, SchNet4AIM can aid the identification of the driving process behind intricate chemical processes at a negligible computational cost when compared to conventional quantum-mechanical calculations.

At low temperature (300 K), the $CO_2$ molecule remains trapped within the supramolecular cage, enabling the former to establish prominent chemical contacts with the different moieties of 13P. For instance, SchNet4AIM predicts clear-cut outbursts (up to 0.10 electrons) in the electron delocalization between the guest molecule and one of the $NH_2$ groups, $\delta(CO_2,NH_2^1)$, as shown in blue in Fig. 5B (jj), which suggest the formation of highly directional interactions. This is, in fact, corroborated by the XCAI analysis of the predictions, which reveals that these contacts are driven by the local N–O and H–O interactions between the $NH_2$ and $CO_2$ moieties. This can be clearly seen in Supplementary Figs. 47 and 48. A visual inspection of the $\delta(CO_2,NH_2)$ (Supplementary Fig. 19) allows to sport discrete binding events, where the $NH_2$ groups take an active role in the stabilization of the ligand within the binding pocket. In fact, there is an ideal agreement between the position of the local maxima (e.g., 1318 fs for $\delta(CO_2,NH_2^1)$) in the latter and the geometries sampled throughout the simulation, as shown in Fig. 5A. Analogous trends are found for the $CO_2$–Ph and $CO_2$–OH delocalized electron counts (Supplementary Fig. 19), evidencing the promiscuity of the guest molecule at establishing additional contacts, such as $\pi$-$\pi$ or dipole-dipole interactions. However, unlike in the case of the $CO_2$–$NH_2$ contacts, the explained SchNet4AIM predictions gathered in Supplementary Figs. 49 and 50 suggest that the $CO_2$–OH bonds are entirely driven by O–O dipole–dipole interactions. This result is in perfect agreement with chemical intuition since the proton-like H atoms of the OH groups, embedded in a strong HB network, have a diminished ability to share electrons with the nearby $CO_2$ molecule.

Obtaining similar XCAI insights from plain geometrical features, comprised in Supplementary Fig. 23, is considerably more challenging, as evidenced by the correlation plots, shown in Fig. 5B (further details

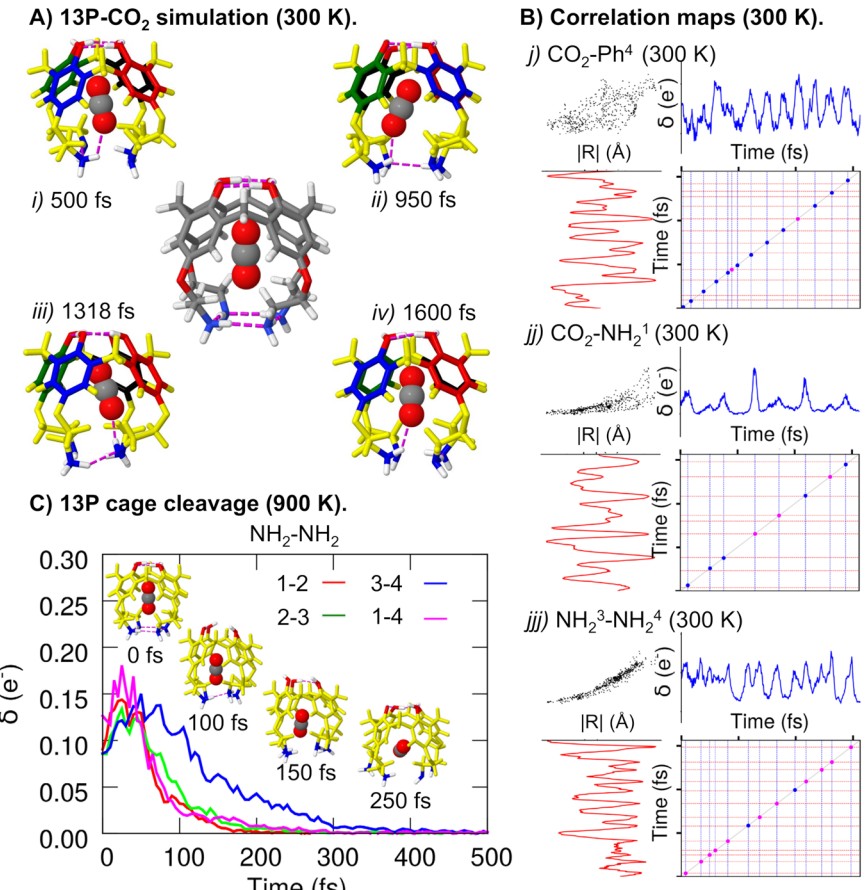

**A) 13P-CO₂ simulation (300 K).**

*i)* 500 fs   *ii)* 950 fs

*iii)* 1318 fs   *iv)* 1600 fs

**C) 13P cage cleavage (900 K).**

**B) Correlation maps (300 K).**

*j)* $CO_2$-$Ph^4$ (300 K)

*jj)* $CO_2$-$NH_2^1$ (300 K)

*jjj)* $NH_2^3$-$NH_2^4$ (300 K)

**Fig. 5 | 13P-CO₂ complexation and guest release. a** 13P-CO₂ system at 500 (i), 950 (ii), 1318 (iii), and 1600 (iv) fs throughout the 300 K simulation, corresponding to some of the local maxima in the electron delocalization ($\delta$) between $CO_2$ and $NH_2$, $\delta(CO_2,NH_2)$. **b** Correlation maps between the electronic and geometrical descriptors (i.e., the distance between the centers of mass, |R|) for the $CO_2$-$Ph^4$ (*j*), $CO_2$-$NH_2^1$ (*jj*), and $NH_2^3$-$NH_2^4$ (*jjj*) contacts throughout the 300 K simulation. Blue dots indicate those binding events exclusively predicted by SchNet4AIM, whereas magenta ones show contacts simultaneously (±10 fs) estimated by the geometrical and electronic metrics. Each tick corresponds to 1000 fs. **c** 13P-CO₂ system at 0, 100, 150, and 250 fs throughout the 900 K simulation, along with the SchNet4AIM computed electron delocalization between neighboring $NH_2$ moieties, $\delta(NH_2-NH_2)$. For the electronic metrics, reported in electrons (e⁻), the raw and bin-averaged data are shown. Time is reported in femtoseconds (fs). Source data are provided as a Source Data file.

on the rationale behind the correlation maps can be found in Supplementary Note 15). The latter shows the binding events, characterized by local maxima or minima in the evolution of the $\delta$ and |R| values, respectively, throughout the simulation. The agreement between electronic and geometrical predictions (shown in magenta) increases as one moves from top to bottom, going from almost null to high exponential correlations between both descriptors. The inability of plain distances to capture orientation-driven contacts, coupled with the information loss attributed to a point-to-point description, becomes particularly detrimental for large groups with a considerable amount of non-H atoms. The combination of these factors, along with the short-sightedness of geometrical features in the description of chemical interactions, explains the drop in performance of the latter at detecting the $NH_2$–$NH_2$ > $CO_2$–$NH_2$ > $CO_2$–Ph contacts.

Increasing the temperature to 900 K promotes larger perturbations of the equilibrium geometry of the 13P skeleton, resulting in more dramatic changes in the electron delocalization at the top and bottom of the cage, as monitored by the $\delta(OH,OH)$ and $\delta(NH_2,NH_2)$ metrics (Supplementary Fig. 20). The thermal energy available at this temperature is more than enough to partially disrupt the HB networks holding the supramolecular cage, something which is particularly detrimental for the highly flexible bottom scaffold of 13P. Such a phenomenon is uniquely reflected in the XCAI $\delta(NH_2,NH_2)$ predictions (Fig. 5C): the geometrical distortion starts to disrupt these interactions at about $t = 25$ fs, and by $t = 100$ fs these have dropped to half their

starting value. The slightly longer-lasting $NH_2^3$–$NH_2^4$ contact (shown in blue) is further weakened at $t = 250$ fs, resulting in the full cleavage of the cage, as corroborated by the trajectories.

From this point on, the alkyl-amine scaffolds get further apart from each other, leading to null electron delocalizations. After the $(NH_2)_4$ HB network cleavage, the ligand gets progressively unbounded from the Calix, leading to the full dissociation of the system at $t \approx 1500$ fs. Within this process, some weak interactions blossom with the different moieties of the receptor, as evidenced by the evolution of their delocalized electron counts. These are primarily driven by multiple $CO_2$ contacts with the unsaturated backbone of the receptor, as evidenced from the bottom panel of Fig. 6 and lasting up to $t \approx 1000$ fs. However, some sporadic interactions are additionally observed within this time window: for instance, at $t = 750$ fs, the $\delta$ metrics predict a prominent contact with one of the $NH_2$ groups which, according to the SchNet4AIM explanations, emerge from the simultaneous formation of N–O and H–O interactions with one of the O atoms in $CO_2$, see Supplementary Fig. 51. This event is actually preceded by a more subtle, though noticeable, spike in the $\delta(CO_2,O)$ electron delocalization which arises from the spatial approximation of the ligand to the O linker of the same arm of the receptor, being completely driven by the O–O interaction between both moieties, see Supplementary Fig. 52. These observations, also found for the remaining correlation plots (Supplementary Figs. 32–37), suggest that the here proposed group electron delocalization is a much more robust and trustworthy metric

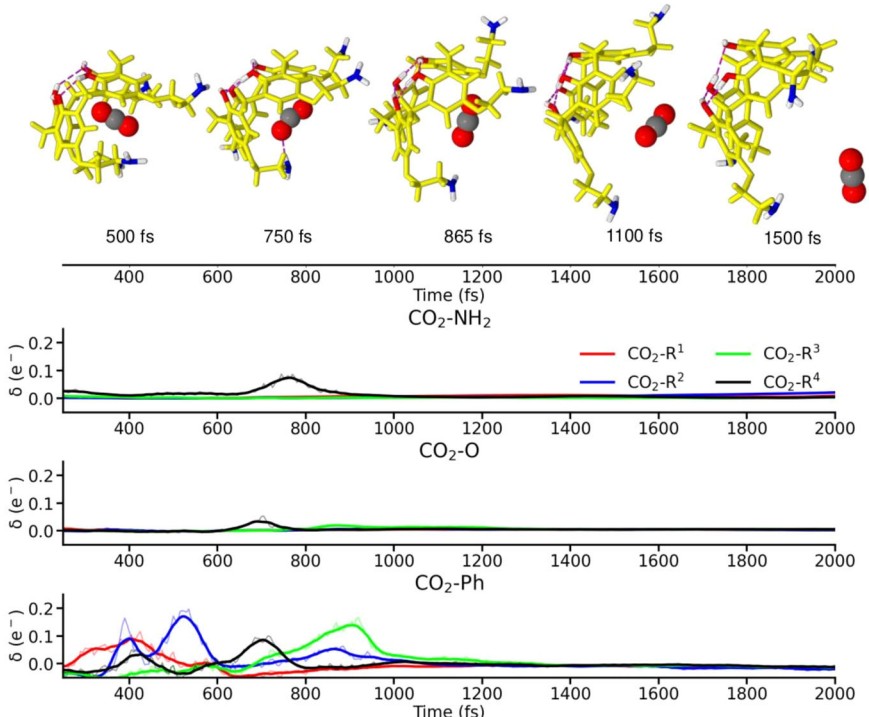

**Fig. 6 | Chemical interactions between $CO_2$ and 13P after the cage cleavage.** Evolution of the SchNet4AIM predicted electron delocalization ($\delta$) between $CO_2$ and the phenyl (Ph), oxo (O), and amino ($NH_2$) moieties of 13P throughout the 13P-$CO_2$ 900 K simulation and after the cage rupture event. Time is reported in femtoseconds (fs). The labels $R^1$, $R^2$, $R^3$, and $R^4$ denote the position of the $NH_2$, O, and Ph moieties in 13P using the same numbering and color code as that shown in Fig. 5A. Source data are provided as a Source Data file.

of binding events of very different nature, even in those scenarios where geometrical features fail. Finally, it is noteworthy that these group properties, which have proven to be valuable in the study of supramolecular systems, have not been explicitly learned during the training. Instead, the latter results from the addition of the physically rigorous SchNet4AIM outputs. Altogether, the interpretability of the SchNet4AIM local predictions, even in unexplored domains, highlights the XCAI abilities of the resultant models. It should also be noted that the findings discussed in this section are in consonance with the results afforded by conventional quantum chemical methods. This is gathered in Supplementary Note 16, which shows a direct comparison of the SchNet4AIM-predicted and quantum-chemically computed QTAIM electron delocalizations throughout the 13P–$CO_2$ 900 K simulation. The excellent correspondence between these two in the estimation of chemical contacts and the electron delocalization accompanying the latter further evidences the reliability and generalizability of the here shown SchNet4AIM models. In fact, remarkably good MAE and RMSE errors, in the range of $10^{-3}$–$10^{-2}$ electrons (see Supplementary Table 24), were observed in the estimation of the group electron delocalizations, even in such intricate extrapolation scenarios.

## Discussion

Achieving a physically coherent picture of complex chemical phenomena is often only possible under the magnifying glass of computationally expensive techniques. In this framework, the use of atomic descriptors, such as those arising within the quantum theory of atoms in molecules and related methodologies, has been limited to the study of small molecules while being unfeasible for the larger systems used in most modern chemical research. Machine Learning models alleviate this problem at the expense of interpretability so that the resulting highly accurate "black box" models are only capable of providing quantitative values. Additionally, accurately condensing chemical information into a machine-readable format still remains one of the

main challenges in its application to chemistry. SchNet overcomes this problem through continuous convolutional filter-generating models, avoiding the tedious task of finely tuning the hand-crafted features while showing state-of-the-art performance in the estimation of molecular properties.

In this work, we have modified the SchNet architecture to enable the prediction of atomic and interatomic quantum chemical quantities. Our approach, SchNet4AIM, has been put to the test with a collection of energetic and electronic terms, including one-body, such as the kinetic energy or the atomic charge, and pair-wise properties, such as the delocalization index or the interatomic energy. SchNet4AIM yields not only accurate but also physically consistent predictions, crucial to reconstructing the expectation values of molecular observables. As such, and following Coulson's maxim, our approach provides valuable chemical insights without compromising the prediction accuracy, being a clear example of an explainable chemical artificial intelligence (XCAI) model. Moreover, such a feature emerges naturally from the use of physically rigorous local quantum chemical properties, which in turn results in intrinsically explainable models. In interpolation regimes, SchNet4AIM outperforms previous approaches[44], increasing the prediction accuracy by more than an order of magnitude while reducing the number of reference data points required for model creation. Besides that, SchNet4AIM has remarkable extrapolation abilities, yielding fairly robust and reliable predictions far away from the sampling domains used for training and contributing to the development of quasi-transferable, multi-purpose ML models. In this way, we have shown how ML can be used for the prediction of atomic or group quantities with great success, contributing to the extension of the physical rigor of quantum chemistry to previously prohibitive computationally intensive realms such as molecular dynamics simulations.

Using the $CO_2$ intake by an aromatic macrocycle as a proof of concept model, we demonstrated how one can achieve a chemical

understanding of complexation in non-covalently bonded supramolecules on extended time scales, even when using SchNet4AIM models trained on small-size, equilibrium molecules of a much simpler chemical space. Moreover, our findings demonstrate that the group electron delocalization, directly obtained from the aggregation of the SchNet4AIM pairwise outputs, is a valuable indicator for a manifold of binding events, even in those scenarios where geometrical features fail, such as in orientation-driven bonding. Besides predictable contacts, XCAI models can track very subtle phenomena (e.g., a hydrogen-bond network rupture) with little to no impact on the total energy of the system. In this way, SchNet4AIM has been shown to provide interpretable and robust predictions, even when employed in completely different energy landscapes and chemical spaces than those used for the training. Additionally, we have shown how the interpretation of SchNet4AIM predictions allows to easily identify the dominant pairwise interactions driving the aforementioned binding events. This, which can be generalized to other domains with the aid of the right real-space tools, opens the door to countless applications, such as the in silico design of tailor-made molecules and materials or the in-depth understating of supramolecular complexation phenomena, to name a few. In the context of XCAI, two key challenges in achieving explainability can be identified. Firstly, the rapid growth of atomic pairs ($\approx M^2$) with the size of the system ($M$), poses difficulty in discerning the most dominant pairwise interactions, especially in complex supramolecular systems with a large number of groups. Secondly, the current limitations of our XCAI implementation restrict the explanations of how the group or molecular properties are influenced by their local components, while the latter cannot be easily traced back to the starting physical variables. However, employing SchNet4AIM in conjunction with extrinsically explainable approaches may address this limitation.

All things considered, our findings point out that the synergy between Quantum Chemical Topology and Machine Learning will likely crystallize in computational tools with many potential applications. Indeed, the development of explainable and robust SchNet4AIM models is expected to be mostly limited by the quality of the underlying reference data, which is often remarkably expensive to compute. Furthermore, fine-tuning of the SchNet representation parameters along with the implementation of active learning protocols could further improve the net performance of the models, something to be explored in the near future. In this way, we have shown how the combination of physically rigorous theories and advanced Deep Learning architectures can contribute to the whitening of AI models. Our proposal is able to provide not just numbers but chemically interpretable knowledge. Thus, the computational bottleneck that prevented the use of rigorous chemical descriptors to understand real-life applications begins to be broken.

## Methods
### Quantum chemical topology
The structure of the electron density, ρ, like that of any scalar field, is stored in the number and type of the critical points of its associated gradient field, ∇ρ. The attraction basins of the local maxima of such a field induce a topology in $R^3$. Since, thanks to Kato's cusp condition[92], each atomic nucleus is homeomorphic to such a maximum, the space is partitioned, in general, into as many 3D regions as nuclei. The electron density thus provides an exhaustive decomposition of $R^3$ into so-called atomic basins, Ω, and for an $M$ nuclei system, $R^3 = \bigcup_A^M \Omega_A$. This is the starting point of QTAIM[12]. Once $R^3$ is divided this way, every global expectation value will become a sum of basin or domain contributions. In the case of a one- ($\hat{O}$) and two-electron ($\hat{G}$) operators[19]

$$\langle \hat{O} \rangle = \sum_A^M \langle \hat{O} \rangle_A, \quad \langle \hat{G} \rangle = \sum_A^M \sum_{B>A}^M \langle \hat{G} \rangle_{AB}, \quad (1)$$

respectively. Experience has also shown that the above domain expectation values are transferable in the chemical sense: the atomic observables of functional groups in chemically similar environments are also similar. For the sake of clarity, in what follows we will just consider a minimal subset of these domain observables. The total electron count ($N$) can be obviously decomposed into domain contributions as

$$N = \int_{R^3} \rho(\mathbf{r}) d\mathbf{r} = \sum_A^M \int_{\Omega_A} \rho(\mathbf{r}) d\mathbf{r} = \sum_A^M N_A, \quad (2)$$

so that atomic charges $q_A = Z_A - N_A$, so often employed to rationalize a wide variety of chemical phenomena[93–95], are thus defined in an orbital invariant manner. Additional electron counting descriptors can be formulated from the statistical analysis of population-related operators. For instance, the variance ($\sigma^2$) of the electron population in a given region is one of the most clear measures of spatial electron localization. It is usually disguised as a so-called localization index ($\lambda$):

$$\lambda(A) = N_A - \sigma_A^2. \quad (3)$$

Obviously, the spread of electrons sensed by $\sigma_A^2$ implies a correlation of the population of different regions that can be quantified by the appropriate covariance, $\sigma_{A,B}$, which is typically known as the delocalization index, $\delta(A, B)$:

$$\delta(A,B) = -2\sigma_{A,B}, \quad (4)$$

the latter are invariant generalizations of the classical two-center bond order[96], collapsing onto the Wiberg-Mayer index[97,98] when this formalism is translated into the traditional orbital language. $\lambda$ and $\delta$ rest on the irreducible part of the two-particle reduced-density matrix (2RDM), being thus sensitive to electron correlation effects. Further order cumulants[99] of the statistical distribution of electron populations have been used to provide multi-center bond orders. Since the variance of the total molecular electron count $N$ vanishes, its localized and delocalized contributions satisfy a sum rule, which is chemically interpreted as the constancy of the total electron population after it is dissected into localized and delocalized contributions:

$$N = \sum_A^M \lambda(A) + \frac{1}{2} \sum_A^M \sum_B^M \delta(A,B). \quad (5)$$

Turning to energetics, the interacting quantum atoms (IQA) approach[14] simply decomposes the expectation value of every term in the standard Coulomb Hamiltonian, writing the energy as a sum of intra-atomic ($E_{\text{intra}}$) and pairwise-additive interaction terms ($E_{\text{inter}}$):

$$E = \sum_A^M E_{\text{intra}}^A + \sum_A^M \sum_{B>A}^M E_{\text{inter}}^{A,B}. \quad (6)$$

This one-body or self-energies group all the kinetic ($T$) and potential ($V$) atomic terms that persist in the dissociation limit,

$$E_{\text{intra}}^A = T^A + V_{ee}^A + V_{Ne}^A, \quad (7)$$

where $V_{ee}^A$ and $V_{Ne}^A$ are intra-atomic electron-electron repulsions and electron–nucleus attraction potential energies, respectively. On the other hand, the inter-basin electronic and nuclear repulsion, along with the electron-nucleus mutually attractive terms, comprise the total interaction energy between any two basins:

$$E_{\text{inter}}^{A,B} = V_{ee}^{A,B} + V_{NN}^{A,B} + V_{Ne}^{A,B} + V_{eN}^{A,B}. \quad (8)$$

It should be noted that the electron-nucleus interaction between atoms $A$ and $B$ arises from the mutual attraction between the electron

density in one of the basins and the nuclear attractor of the other one, yielding two well-defined contributions, namely $V_{Ne}^{A,B}$ and $V_{eN}^{A,B}$.

## SchNet4AIM: model construction and training

The SchNetPack hyperparameters used in this work can be found in Supplementary Note 6. SchNet4AIM was tested against two different databases, decomposed in standard training–testing–validation splits, as detailed in Supplementary Note 8. The first of these comprises a set of QTAIM electron metrics of C, H, O, and N containing neutral molecules in their equilibrium or near equilibrium geometries. For this purpose, a 10% random subsample of a previously developed database[44], gathering neutral and singlet-spin structures of the CHON equilibrium space, was employed. For each molecule, the QTAIM atomic charges (M06-2X/def2-TZVP) are already provided, whereas single-point calculations were computed at the same level of theory to obtain the remaining electronic descriptors ($\lambda$ and $\delta$).

On the other hand, a second database was built to explore the applicability of SchNet4AIM in the realm of energy prediction, relying on IQA energetic partitioning. Considering the large computational cost attributed to the latter, we decided to employ a set of smaller and less chemically complex systems than those used in the aforementioned QTAIM database. For such a purpose, ab initio molecular dynamics (BLYP-D3/6-31G(d,p), 300 K) were performed on a collection of gas-phase water clusters $(H_2O)_n$ with $n = [1–6]$, gathered from the Cambridge database[100]. Structures were extracted every 10 fs (time-step = 1.0 fs) to roughly sample the potential energy surface, resulting in a total of 1016 geometries. Single-point calculations (B3LYP/6-31G(d,p)) were run on the latter to obtain the corresponding wavefunctions from which the IQA partitioning of the total electronic energy was achieved. For the sake of computational time, medium-density integration grids were used throughout. Thus, lower performances (when compared to the ML estimation of common chemical properties) are generally expected when dealing with this reference data. Details on the training protocol used for the training of SchNet4AIM in combination with both databases can be found in Supplementary Note 6. Additionally, the format used for the construction of SchNet4AIM databases is specified in Supplementary Note 3.

In order to show the applicability of the SchNet4AIM predictions as valuable indicators of chemical interactions in dynamic scenarios, atomistic molecular dynamics calculations of a selection of supramolecular systems were performed in the gas phase, starting from the optimized geometries available in the literature[72] at the HF-3C level of theory as implemented in the ORCA quantum chemistry package[101]. Time-steps in the range of 1–1.25 fs were used throughout for a varying number of steps ranging from 2500 to 3500. The temperature was kept constant with the aid of a Berendsen thermostat (set at 300 or 900 K) with a coupling constant of 5–10 fs. As geometrical features, the distance between the center of mass was employed.

The data gathered in the figures will be, in general, reported in atomic units (a.u. or electrons) owing to the large nominal values for some of the here studied quantities. However, and for the sake of simplicity, the more commonly employed unit of kcal mol$^{-1}$ (1 a.u. $\approx$ 627.5 kcal mol$^{-1}$) will be used to refer to the energetic quantities throughout the discussions. Further information about the computational resources and codes employed in this work can be found in Supplementary Note 4. All molecular representations were rendered with Jmol[102].

## Reporting summary

Further information on research design is available in the Nature Portfolio Reporting Summary linked to this article.

## Data availability

The data generated in this study, including details about algorithms, computational calculations, database creation, ML training, and additional performance tests, are provided in the Supplementary Information. Additionally, the actual databases used to train our models are available at the SchNet4AIM GitHub repository. Source data are provided with this paper[103]. Source data are provided with this paper.

## Code availability

The code and pre-trained SchNet4AIM models will be available, online, at the SchNet4AIM GitHub repository[104] (https://github.com/m-gallegos/SchNet4AIM). The Gaussian and ORCA suites employed for basic quantum chemical calculations are available from Refs. [105] and [101], respectively. On the other hand, the PROMOLDEN and AIMAll codes used to run the QTAIM/IQA calculations are gathered in Refs. [106] and [107] respectively.

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

## Acknowledgements

The authors kindly acknowledge the Spanish MICIU (No. PID2021-122763NB-I00 to A.M.P. and M.G.) for financial support. Additionally, M.G. especially thanks the Spanish MICIU for the predoctoral FPU grant (Nos. FPU19/02903 and EST22/00100 to M.G.).

## Author contributions

All authors contributed equally to the writing of this paper. M.G., V.V.-G., and I.P. performed the computational studies and algorithmic development. I.P., A.M.P., and A.T. supervised this work and designed the main workflow and analysis. All authors have given approval to the final version of this paper.

## Competing interests

The authors declare no competing interests.
