## [Peer Review File · Nature Communications]

Explainable Chemical Artificial Intelligence From Accurate Machine Learning of Real-Space Chemical DescriptorsREVIEWER COMMENTS

Reviewer #1 (Remarks to the Author):

In their paper, "Explainable Chemical Artificial Intelligence: Accurate Machine Learning of Real-Space Chemical Descriptors", the authors develop a machine learning (ML) model based on the SchNet architecture that addresses the trade-off between explainability and accuracy in ML computational chemistry, enabling the prediction of various real-space quantities with improved accuracy and speed.

While their work is interesting and a valuable addition to the existing body of literature, I do not believe it meets the criteria for a publication in Nature Communications. While it is clear that explainable artificial intelligence (XAI) is still in its early stages of development within the field of computational chemistry, this work represents a significant step forward. However, it appears to be tailored to a relatively specific audience, as it primarily serves as a proof of concept for XAI. It would have been more impactful to include a demonstration of XAI in a key application where XAI can significantly enhance the prediction of chemical information. See also item 4 in the annotated PDF.

Their work, however, certainly merits publication at a different venue for a more specialized audience.

Besides my main criticism above, I have additional technical questions which should be considered before proceeding further. I have provided my comments and questions below, which can also be found in the annotated PDF.

1: There are several "physically sound ways" of partitioning an interacting quantum system in the context of molecules. Why do you highlight this one and do not refer to others?

2: Same as 1, there are other methods to partition a molecular system into (interacting) fragments such as <https://doi.org/10.1021/jp066449h> and <https://doi.org/10.1103/PhysRevA.82.024501> and others. In my opinion, this part of the manuscript is biased and needs to be extended to account for other related work. This would make the manuscript accessible for a broader audience.

3: I suggest to include more references here to justify the word "outburst". There are certainly more works on this topic in the recent literature. Also review articles could be cited.

4: My primary criticism:

While it's clear that explainable artificial intelligence (XAI) is still in its early stages of development within the field of computational chemistry, this work represents a significant step forward. However, it appears to be tailored to a relatively specific audience, as it primarily serves as a proof of concept for XAI. It would have been more impactful to include a demonstration of XAI in a key application, or at the very least, demonstrate aspects of a key application where XAI can significantly enhance the prediction of chemical information.

5: Equation (8):

Can you explain why you distinguish between $V_{\{Ne\}^{\{A,B\}}}$ and $V_{\{eN\}^{\{A,B\}}}$? This might just be a matter of notation, but I'm wondering why you explicitly include two terms instead of just

summarizing this energy contribution in one term.

6: The premise of the authors' ML model remains unclear. Could you clarify whether the goal is to develop a single model that is transferable across a large chemical space or do you aim to build models suitable to a subset of that space?

It is also unclear here on which database the ML model is trained and tested on. Here, a water cluster database is mentioned, but no details are given. Is the water cluster database a subset of the QTAIM database of Supplementary Note 8?

What is the training and testing protocol for the results shown in Fig. 2?

Furthermore, the computational efficiency of the ML model is a key focus of this study. It would be essential to include a comparison of computational times between the conventional method and the ML approach developed by the author.

7: This is a very interesting test of transferability of the authors ML model. I might be misunderstanding this part of the manuscript, but it would be highly desirable to compare the ML model prediction with a baseline provided by quantum chemical methods (for example the predicted delocalization Δ in Fig. 5c). If this is not feasible, this should be stated, and a different metric for performance/accuracy should be provided.

8: See my comment related to computational timings in 6. Also the cost for training should be accounted for in the discussion.

Reviewer #2 (Remarks to the Author):

This paper is concerned with Explainable Chemical Artificial Intelligence: Accurate Machine Learning of Real-Space Chemical Descriptors. This is an important and timely research topic, and is likely to be of general interest to the readership of Nature Communication.

The authors present the authors developed SchNet4AIM, a SchNet-based architecture capable of dealing with local one-body (atomic) and two-body (interatomic) descriptors. The performance of SchNet4AIM could be tested by predicting a wide collection of real-space quantities ranging from atomic charges and delocalization indices to pairwise interaction energies. The accuracy and speed of SchNet4AIM breaks the bottleneck that has prevented the use of real-space chemical descriptors in complex systems. The authors show that the group delocalization indices, arising from their physically rigorous atomistic predictions, provide reliable indicators of supramolecular binding events, paving the way towards Explainable Chemical Artificial Intelligence (XCAI) models. The research in this paper would benefit for addressing the dilemma between explainability and accuracy.

It is my recommendation that the paper should be published in the Nature Communication.

Reviewer #3 (Remarks to the Author):

Review: Explainable Chemical Artificial Intelligence: Accurate Machine Learning of 2 Real-Space Chemical Descriptors

This paper outlines an approach to combine Quantum Chemical Topology (QCT) with models machine learning models in the form of SchNet. The new model the authors name SchNet4AIM is trained initially on water clusters and its interpolation abilities are tested. The abilities are shown to be in line with previous results from others achieving 1 – 2 kcal per mole MAE. The authors then extend the models to a wider set of atoms (H,C,N and O). The models are trained on a portion of this data and then applied in an extrapolation setting to predict quantities over a reaction trajectory. The results show that the models can make very good approximations to quantum chemical calculations even in this extrapolation regime. This is further explored by application to CO₂ capture and release by the cyclic oligomer calix[4]arene. The authors show that chemical insights can be gained by using the model to predict electron delocalization.

The results here are very insightful and relevant to the topic of machine learning potentials/neural potentials. These results show a promising direction.

I recommend that this article is published after the following minor comments are addressed.

Line 81: The author's comment on the problem of the growing intricacy of the problems ML is dealing with and that there is a scarcity of uncertainty metrics. I don't see any reference to uncertainties for the model presented here. Could the author's comment on any attempts they made to include uncertainty (ensembles for example such as in the ANI networks)? If not perhaps they could suggest avenues for exploration following their work?

Line 106: The phrase very relevant properties is quite nebulous could this be clarified please?

Paragraph under equation 4: You specify the abbreviation 2RDM which I take to be the second order reduced density matrix, however, I don't believe this is specified anywhere. Please include this specification

Figure 1: The pooling layer title appears to be crossed through

Line 141: The author mention that it is possible to envisage many ways to construct atomic pair descriptor vectors. The authors clarify their chosen method, but I wonder if the author can provide any further comments on other options they tried whilst developing the model? This would be very useful to the community. I think this could be added to SI and hence not impact the text greatly in the main manuscript.

Line 205: The author's generate several models during the course of the paper, but the only comparison in terms of the amount of data to train the model Vs. the models accuracy is a comparison some of the authors previous neural network model, as far as I can see. Could they comment upon the effect of training data set size on these models?

The GitHub link seems to lead to a 404 gateway meaning we can not see the code or pre-trained models in the review <https://github.com/mgallegos/SchNet4AIM>.

It is not clear from the data availability statement whether the authors intend to make the training data available in full. I would urge the author to do so to enable the community to build on the work in addition to making the work highly transparent.

Typos and references

References 542 and 561 seem to be repeated.

Line 249 "on.The" missing space

Reviewer #4 (Remarks to the Author):

- The authors propose and evaluate an interesting modification of SchNet with 1P and 2P descriptors. They demonstrate performance in global and local predictions tasks, extrapolation and interpretability tasks.
- The results are promising and the provided evaluation set ups are convincing.
- The paper is well-written and provides just the right amount of detail for a non QTAIM expert to understand the fundamental points. Also, I notice the SI is well documented and it looks like the final code implementation will be very useful.

Overall, I recommend the work for publication and suggest the authors to address these considerations:

1. The motivation and method for SchNet4AIM are clearly explained. However, it took me a careful review to identify 1) main modifications compared to ScNet and 2) explainability piece. I think the authors may benefit by modifying Figure 1 to highlight the main modifications of the ScNet architecture (1P and 2P descriptors, different outputs), as you are currently doing with the "pooling layer". Also, you could identify the "explainable" elements in the schematic.

2. The model you are proposing is intrinsically explainable -- as the model outputs can be used for explainability. This is nicely demonstrated in the final case study. However, this is not clearly stated in the text. I suggest you mentioned this and comment you are not using an extrinsic interpretability or proving technique to provide explanations. A few sentences explaining how the model provides interpretable outputs and how to use these predictions, may be enumerating some potential use cases, can be useful for non-expert readers.

3. The extrapolation task shows very promising results and I agree with the authors about the predictive power of 1P and 2P features. What are the limitations to this extrapolation capabilities? The authors mentioned that larger and more complex systems will be more challenging, but is there any quantification or observation of these based on the modeled systems / case studies? How do the authors balance / regularize the model complexity in such cases? I saw a brief mention of it in the SI, but I would appreciate clear discussion in the main manuscript.

4. Also, in the extrapolation task, the authors used 10% of the dataset for training, is that enough to be defined as extrapolation? For me this is a few shot learning task, but I understand this can be defined different in the literature for the specific problem. What happens when training with less data?

5. What are the model limitations for further use in the SchNet applications?

6. Does the training datasets require any specific features compared to traditional datasets used to train SchNet in multiple applications?

7. Is permutation invariance in 2P features enough or could the model be further expanded with translational or rotational invariance? I had to look at the original SchNet paper to understand this piece so it might be good clarify this.

8. What are potential model improvements?

9. Regarding explanations, I think the interpretability case study is great illustration of the model capabilities. What are some limitations in terms of predictive power and explainability? Can those be discussed more broadly in the paper along with potential explainability use cases?

10. Model explanations as outputs are very useful, but are there limitations to them? For example, defining more specific / macro interpretability physical quantities that are not as clear connected to atomistic interaction? Can the features / learned representations be interpreted too? Are extrinsic explainability techniques recommended or necessary?

REVIEWER COMMENTS

Reviewer #1 (Remarks to the Author):

In their paper, "Explainable Chemical Artificial Intelligence: Accurate Machine Learning of Real-Space Chemical Descriptors", the authors develop a machine learning (ML) model based on the SchNet architecture that addresses the trade-off between explainability and accuracy in ML computational chemistry, enabling the prediction of various real-space quantities with improved accuracy and speed. While their work is interesting and a valuable addition to the existing body of literature, I do not believe it meets the criteria for a publication in Nature Communications. While it is clear that explainable artificial intelligence (XAI) is still in its early stages of development within the field of computational chemistry, this work represents a significant step forward. However, it appears to be tailored to a relatively specific audience, as it primarily serves as a proof of concept for XAI. It would have been more impactful to include a demonstration of XAI in a key application where XAI can significantly enhance the prediction of chemical information. See also item 4 in the annotated PDF. Their work, however, certainly merits publication at a different venue for a more specialized audience. Besides my main criticism above, I have additional technical questions which should be considered before proceeding further. I have provided my comments and questions below, which can also be found in the annotated PDF.

We thank the reviewer for their careful evaluation and review of our work, which will contribute to improving its overall quality while making it more accessible to a broader (non-specific) audience. In the upcoming pages, we provide a point-by-point response to the comments and suggestions raised by the reviewer, while detailing how these have been addressed.

1. 1: There are several "physically sound ways" of partitioning an interacting quantum system in the context of molecules. Why do you highlight this one and do not refer to others?

As very conveniently stressed by the reviewer, a chemical system, along with its properties, accepts multiple partitions, being the Atoms in Molecules (AIM) approach one of them. In fact, a lot of work has been done in this regard in the last decades, which has resulted in the proliferation of countless population analysis tools and partitioning schemes. In this context, much interest has sparked in the study and decomposition of molecular electron density, owing to its pivotal role for both theoreticians and experimentalists. As such, many methods have been proposed in this regard, including the iterative Hirshfeld method (<https://doi.org/10.1063/1.2715563>), DDEC6 (<https://doi.org/10.1039/C6RA04656H>), Becke's integration approach (<https://doi.org/10.1063/1.454033>), the Bader's Atom in Molecules methodology used in this work (*Bader, R. F. W. (1990). Atoms in Molecules: a Quantum Theory. Oxford University Press*) along with the ubiquitous Mulliken (<https://doi.org/10.1063/1.1740588>) and Löwdin (*J. Chem. Phys.* 21, 374–375, 1955) analyses, to name just a few.

An even larger variety of tools has been proposed to express the total molecular energy as a collection of additive terms, being Symmetry Adapted Perturbation Theory (SAPT) (<https://doi.org/10.1080/00268977900101601>, <https://doi.org/10.1021/acs.jctc.1c00344>) and the famous Energy Decomposition Analysis (EDA) (<https://doi.org/10.1002/wcms.1345>) along with its related Activation Strain Model, also known as Distortion/Interaction approach, (<https://doi.org/10.1039/D0CP04016A>) two of the most widespread strategies.

However, these and similar approximations are unfortunately bound to provide an arbitrary system description due to the inherent bias accompanying the dependence on elusive references. The vulnerabilities and inconveniences of the latter have been exhaustively explored and criticized in the past (see, for instance, <https://doi.org/10.1039/D1CP04135E>, <https://doi.org/10.1039/D0CP04016A>). As some of us have recently put forward (<https://doi.org/10.1038/s41467-022-31036-6>), real-space analyses provide a very convenient solution to this problem, paving the way towards a more robust and rigorous picture of chemistry in general, and chemical bonding in particular. In addition to being physically sound, the real space description of an atom opens the door to the highly pursued chemical transferability, the key to developing better generalizable and transferable Machine Learning (ML) models. We have used real-space techniques to obtain local chemical properties without arbitrariness. In conjunction with the transferability, this represents a comprehensive and promising strategy to advance in the context of Explainable Artificial Intelligence (XAI).

1. 2: Same as 1, there are other methods to partition a molecular system into (interacting) fragments such as <https://doi.org/10.1021/jp066449h> and <https://doi.org/10.1103/PhysRevA.82.024501> and others. In my opinion, this part of the manuscript is biased and needs to be extended to account for other related work. This would make the manuscript accessible for a broader audience.

Following the reviewer's suggestions, we have included a wider summary of these and similar approaches. Changes can be found in the main text (page 2, lines 44-49), highlighted in blue.

1. 3: I suggest to include more references here to justify the word "outburst". There are certainly more works on this topic in the recent literature. Also review articles could be cited.

Following the reviewer's suggestions, additional references have been included in the main text to provide a more detailed overview of the outburst of the synergy between AI and scientific research, especially within the field of chemical sciences. Changes can be found, highlighted in blue, on page 3 (line 77-82).

1. 4: My primary criticism: While it's clear that explainable artificial intelligence (XAI) is still in its early stages of development within the field of computational chemistry, this work represents a significant step forward. However, it appears to be tailored to a relatively specific audience, as it primarily serves as a proof of concept for XAI. It would have been more impactful to include a demonstration of XAI in a key application, or at the very least, demonstrate aspects of a key application where XAI can significantly enhance the prediction of chemical information.

Following the suggestion raised by the reviewer, we have decided to extend the discussion on the interpretability of SchNet4AIM predictions. To do so, we have tested our approach in the most challenging scenario discussed here: the 13P-CO₂ complexation phenomena. In this way, we show that analysis of the individual SchNet4AIM predictions quickly identifies the most relevant terms that dominate the behavior of the shared electron counts between any two groups throughout the simulations. See the newly added Supplementary Note 18. For instance, the SchNet4AIM predictions revealed that the O-O interaction dominates the behavior of the electron delocalization between the CO₂ molecule and the nearby OH moieties. Instead, the

more accessible N and H atoms of the NH₂ moieties drive the delocalized electron counts between the latter and the trapped CO₂ molecule, as reflected in Supplementary Fig. 47. As such, our results show that SchNet4AIM allows us to elucidate in a univocal way the pairwise interactions that dominate the previously observed supramolecular binding events evidencing the interpretability and rigor brought by the combination of ML and real-space tools.

More discussion about the explainability and the utility has been added to different sections throughout the main text (which can be found, in blue, in lines 324-328, 333-335, 343-348, 380-382, and 384-385).

1. 5: Equation (8): Can you explain why you distinguish between $V_{\{Ne\}^{A,B}}$ and $V_{\{eN\}^{A,B}}$? This might just be a matter of notation, but I'm wondering why you explicitly include two terms instead of just summarizing this energy contribution in one term.

As pointed out by the reviewer, the total electron-nucleus attractive potential (V_{atr}) between any pair of atoms (A,B) defined in the real space, emerges from the combination of two well-defined components:

- The interaction between the electron density (e) of basin A and the nuclear attractor (N) in basin B ($V_{\{eN\}^{A,B}}$).
- The interaction between the nuclear attractor (N) in basin A and the electron density (e) of basin B. ($V_{\{Ne\}^{A,B}}$)

It is thus easy to see that both terms have the same physical origin (the electron-nucleus interaction), evaluated between the constituents of the two atoms (A, B). Naturally, the total attractive potential is just the sum of these two components: $V_{atr}^{A,B} = V_{\{eN\}^{A,B}} + V_{\{Ne\}^{A,B}}$. The $V_{\{eN\}^{A,B}}$ and $V_{\{Ne\}^{A,B}}$ energy components will generally exhibit widely different values, and thus having access to the raw components to the total potential is often helpful to delve deeper into the nature of the interaction between atoms A and B.

A clarifying sentence regarding this topic has been included in the main text for the sake of readability. Changes can be found, highlighted in blue, in lines 146-148 (page 7).

1. 6: The premise of the authors' ML model remains unclear. Could you clarify whether the goal is to develop a single model that is transferable across a large chemical space or do you aim to build models suitable to a subset of that space?

SchNet4AIM is a general-purpose ML model utilizing the robustness of real-space quantum chemical quantities. It constitutes a step forward toward attaining transferable and explainable models. Using reference-free methods results in the physical robustness of the SchNet4AIM, paving the way toward Explainable Chemical Artificial Intelligence (XCAI). Doing so, we expect that the resultant models could be found useful well beyond the specific frameworks where the underlying reference methodologies were derived in the first place, being applicable across many fields within the scientific community.

In this regard, a couple of clarifying sentences have been included in the main text; changes can be found highlighted in blue on page 4 (lines 113-114) and page 5 (lines 122-123).

It is also unclear here on which database the ML model is trained and tested on. Here, a water cluster database is mentioned, but no details are given. Is the water cluster database a subset of the QTAIM database of Supplementary Note 8?

We thank the reviewer for pointing out that, in the original discussion of the results, the origin of the databases used in this work may not have been clear enough to the general reader. The water-cluster database, much smaller in size than the QTAIM database, comprises the IQA energy decomposition of a total of 1016 water clusters (this was specified in Supplementary Note 8, along with the number of training, testing and validation points in which the latter was split). However, for the sake of clarity and to avoid any potential misunderstanding, further details on the databases used in this work have been included in the manuscript.

Changes can be found, highlighted in blue, in the Method section of the text (page 21-22, lines 469-472, 477-480).

What is the training and testing protocol for the results shown in Fig. 2?

As already mentioned in the previous point, details on the water-cluster IQA database are collected in Supplementary Note 8, whereas the training and testing protocol is thoroughly detailed in Supplementary Note 6. For the sake of clarity, this has been explicitly stated in the main text.

Changes can be found in blue in the Methods section of the manuscript (page 22, lines 489-491).

Furthermore, the computational efficiency of the ML model is a key focus of this study. It would be essential to include a comparison of computational times between the conventional method and the ML approach developed by the author.

Following the reviewer's suggestion, we have thoroughly evaluated the scaling laws of conventional approaches and SchNet4AIM with the size of the system, as gathered in a new section of the supplementary material, namely Supplementary Note 17. Our results show that SchNet4AIM entails a negligible computational cost when compared to that required to obtain the quantum chemical reference data. Moreover, and as expected, the computational speedup increases with the size of the system, leading to factors of up to 10^4 in the limiting case of the 48 atoms system explored so far.

Changes can be found in blue, in Supplementary Note 17 of the supplementary material. Also, a brief discussion has been included in the main text; see page 13 (lines 266-269, in blue).

1. 7: This is a very interesting test of transferability of the authors ML model. I might be misunderstanding this part of the manuscript, but it would be highly desirable to compare the ML model prediction with a baseline provided by quantum chemical methods (for example the predicted delocalization Δ in Fig. 5c). If this is not feasible, this should be stated, and a different metric for performance/accuracy should be provided.

As already inferred by the reviewer, the computational cost of the conventional Quantum Theory of Atoms in Molecules (QTAIM) analysis of the electron density of a system (quantum chemical

method) is extraordinarily large. Furthermore, the latter grows rapidly with the size, number of electrons, and number of basis functions of the system (see Supplementary Note 17 for further details). For this reason, we decided not to explicitly compute the quantum-chemically estimated values in the first place.

However, we understand the concerns raised by the reviewer as a direct comparison with the reference quantum chemical calculations could help us to further prove the reliability and robustness of the SchNet4AIM predictions. Considering this, a set of reference calculations was performed on the 13P-CO2 MD 900 K simulation (corresponding to Fig. 5C mentioned by the reviewer) at the M062X/3-21G level of theory. The smaller 3-21G basis set was employed instead of the def2-TZVP, used to train the underlying SchNet4AIM models, owing to its ability to afford similar QTAIM metrics at a much more reduced computational cost. It should be noted that the latter was selected after carefully evaluating the compromise between computational time and accuracy provided by different basis sets in a benchmarking study (more information can be found in the newly added Supplementary Note 16). Our results show that the SchNet4AIM models are able to afford extraordinarily accurate predictions even in such supramolecular scenarios, far from the training space used in the underlying models. This is clearly evidenced by Supplementary Figs. 42-43 shows an outstanding qualitative agreement and a very close quantitative match between the ML predictions and the true quantum mechanically computed data.

Finally, conventional QTAIM algorithms may struggle to analyze complex electron density fields, especially when being very far from the molecular equilibrium domains, which can result in large integration errors and computational artifacts. In fact, this effect can be shown in the left panel of Supplementary Figs. 42-43, where some frames in the trajectory showed very large integration errors and had to be discarded. In this way, not only does SchNet4AIM afford reliable and transferable outputs, but it also comes particularly handy in analyzing certain out-of-equilibrium structures, which are tricky to compute through conventional approaches.

In view of the relevance of these findings, they have given rise to a new section in the supplementary material (Supplementary Note 16) and have been referred to in the main text (see page 19, lines 393-401, highlighted in blue).

1. 8: See my comment related to computational timings in 6. Also the cost for training should be accounted for in the discussion.

As discussed in comment 1.6, we have performed a detailed analysis of the computational times required by SchNet4AIM and quantum-chemically based approaches to estimate some of the here explored real-space quantities, including the actual training times. (Further details can be found in the newly added Supplementary Note 17, Supplementary Table. 25).

As for the computational time entailed in the predictions, SchNet4AIM is considerably faster than conventional approaches, an effect that (as expected) becomes more pronounced with the size of the system. In fact, speedup factors of up to 10^4 were found throughout, with SchNet4AIM taking 1.88 seconds to compute the QTAIM electron metrics of a 48 atoms system, vs the 20506 seconds required by conventional approaches). On the other hand, moderate training times of 6 hours were required to train one-particle models on about 70 000 local values. Naturally, the cost involved in the training of two-particle models, presented with 800

000 local values, rises, reaching reasonable values of 130-140 hours. We hope these results, and related discussion, clarifies the computational efficiency of SchNet4AIM.

Reviewer #2 (Remarks to the Author):

This paper is concerned with Explainable Chemical Artificial Intelligence: Accurate Machine Learning of Real-Space Chemical Descriptors. This is an important and timely research topic, and is likely to be of general interest to the readership of Nature Communication. The authors present the authors developed SchNet4AIM, a SchNet-based architecture capable of dealing with local one-body (atomic) and two-body (interatomic) descriptors. The performance of SchNet4AIM could be tested by predicting a wide collection of real-space quantities ranging from atomic charges and delocalization indices to pairwise interaction energies. The accuracy and speed of SchNet4AIM breaks the bottleneck that has prevented the use of real-space chemical descriptors in complex systems. The authors show that the group delocalization indices, arising from their physically rigorous atomistic predictions, provide reliable indicators of supramolecular binding events, paving the way towards Explainable Chemical Artificial Intelligence (XCAI) models. The research in this paper would benefit for addressing the dilemma between explain ability and accuracy. It is my recommendation that the paper should be published in the Nature Communication.

2.1 It is my recommendation that the paper should be published in the Nature Communication.

We express our sincere gratitude to the reviewer for their insightful comments and feedback about the quality and suitability of our results.

Reviewer #3 (Remarks to the Author):

This paper outlines an approach to combine Quantum Chemical Topology (QCT) with models machine learning models in the form of SchNet. The new model the authors name SchNet4AIM is trained initially on water clusters and its interpolation abilities are tested. The abilities are shown to be in line with previous results from others achieving 1 – 2 kcal per mole MAE. The authors then extend the models to a wider set of atoms (H,C,N and O). The models are trained on a portion of this data and then applied in an extrapolation setting to predict quantities over a reaction trajectory. The results show that the models can make very good approximations to quantum chemical calculations even in this extrapolation regime. This is further explored by application to CO2 capture and release by the cyclic oligomer calix[4]arene. The authors show that chemical insights can be gained by using the model to predict electron delocalization. The results here are very insightful and relevant to the topic of machine learning potentials/neural potentials. These results show a promising direction. I recommend that this article is published after the following minor comments are addressed.

We gratefully thank the reviewer for their time and effort spent in evaluating our results and the conclusions that can be drawn from this work. Following, we provide a point-by-point response to each of the comments raised by the reviewer.

3.1 Line 81: The author's comment on the problem of the growing intricacy of the problems ML is dealing with and that there is a scarcity of uncertainty metrics. I don't see any reference to uncertainties for the model presented here. Could the author's comment on any attempts they made to include uncertainty (ensembles for example such as in the ANI networks)? If not perhaps they could suggest avenues for exploration following their work?

As mentioned in our manuscript, one of the major criticisms of common ML tools is the lack of proper uncertainty metrics accompanying their predictions. Following the suggestion of the reviewer, we have decided to implement a simple uncertainty estimation of the proof-of-concept SchNet4AIM models used throughout our manuscript. Albeit one can come up with a plethora of different ways to estimate the accuracy of a prediction, for instance, performing statistical analysis on the output of different models targeted at computing the same quantity, we have decided to exploit the rigor governing our reference data. For instance, in the case of the here employed QTAIM metrics, the expected number of electrons of the molecule must be exactly reconstructed. In this way, the net molecular error made by the SchNet4AIM models can be estimated from the evaluation of the accuracy with which the total electron counts add up to their molecular analogs. The latter can be obtained either from the atomic charges (Q) or from the combination of the localized (LI) and delocalized (DI) contributions to the electron population of an atom (N). Doing so allows us to obtain a reliable measure of the uncertainty of the here shown SchNet4AIM models targeted at predicting QTAIM electron metrics.

Following the reviewer's suggestion, this has been implemented in the SchNet4AIM repository, and a detailed explanation has been included in Supplementary Note 2 (which can be found highlighted in blue at the end of such section). Additionally, here we include an example of a typical SchNet4AIM output, showing the predicted QTAIM metrics of a molecule along with the corresponding uncertainty metrics:

ATOMIC CHARGE (Q)

Atom 1 (C) q = +2.281 +- 0.001 : +2.280 to +2.282 electrons.
Atom 2 (O) q = -1.255 +- 0.001 : -1.256 to -1.254 electrons.
Atom 3 (O) q = -1.266 +- 0.001 : -1.267 to -1.265 electrons.
Atom 4 (C) q = -0.028 +- 0.001 : -0.029 to -0.027 electrons.
Atom 5 (C) q = -0.026 +- 0.001 : -0.027 to -0.025 electrons.
Atom 6 (C) q = +0.542 +- 0.001 : +0.541 to +0.543 electrons.
Atom 7 (C) q = +0.058 +- 0.001 : +0.057 to +0.059 electrons.
Atom 8 (C) q = -0.014 +- 0.001 : -0.015 to -0.013 electrons.
Atom 9 (C) q = -0.036 +- 0.001 : -0.037 to -0.035 electrons.

.....
.....

ELECTRON DELOCALIZATION INDEX (DI)

Pair 1 2 (C ,O) DI = +1.263 +- 0.001 : +1.262 to +1.265 electrons.
Pair 1 3 (C ,O) DI = +1.266 +- 0.001 : +1.265 to +1.267 electrons.
Pair 1 4 (C ,C) DI = -0.006 +- 0.001 : -0.007 to -0.005 electrons.
Pair 1 5 (C ,C) DI = -0.004 +- 0.001 : -0.005 to -0.003 electrons.
Pair 1 6 (C ,C) DI = -0.004 +- 0.001 : -0.005 to -0.003 electrons.
Pair 1 7 (C ,C) DI = -0.006 +- 0.001 : -0.007 to -0.005 electrons.
Pair 1 8 (C ,C) DI = -0.005 +- 0.001 : -0.006 to -0.004 electrons.
Pair 1 9 (C ,C) DI = -0.003 +- 0.001 : -0.004 to -0.002 electrons.
Pair 1 10 (C ,C) DI = -0.007 +- 0.001 : -0.008 to -0.006 electrons.
Pair 1 11 (C ,C) DI = -0.005 +- 0.001 : -0.006 to -0.004 electrons.
Pair 1 12 (C ,C) DI = -0.006 +- 0.001 : -0.007 to -0.005 electrons.
Pair 1 13 (C ,H) DI = -0.000 +- 0.001 : -0.001 to +0.001 electrons.

.....
.....

As can be seen, for each atom or atomic pair, the SchNet4AIM predicted local value is given along with a particular estimation error, in this case of 0.001 electrons. The latter sets the uncertainty bounds of the prediction. For instance, the atomic charge of atom 1 is 2.281 electrons, which accounting for the uncertainty limits could well be anywhere between 2.280 and 2.282 electrons. Similarly, the delocalization index between atoms 1 and 3 is expected to have a value in between 1.265 and 1.267 electrons. Notice that both the errors and predictions are rounded up while printing, which explains the slightly different uncertainty bounds that are found for certain local values (for example, the DI (1,2) is between 1.262 and 1.265 electrons, which would correspond to an error bound slightly above 0.001).

3.2 Line 106: The phrase very relevant properties is quite nebulous could this be clarified please?

We totally agree with the reviewer's comment; our original expression was somewhat ill-defined and has been updated for the sake of clarity and rigor. Changes are highlighted in blue in line 117-118 of the manuscript (page 4).

3.3 Paragraph under equation 4: You specify the abbreviation 2RDM which I take to be the second order reduced density matrix, however, I don't believe this is specified anywhere. Please include this specification.

The reviewer is right, we forgot to explicitly indicate the meaning of the 2-particle Reduced-Density Matrix (2RDM) in the manuscript. This has now been included in the text. Changes can be found in blue on page 6 of the manuscript.

3.4 Figure 1: The pooling layer title appears to be crossed through.

There was a graphical typo in the original figure. The latter has been fixed; accordingly, changes can be found in Fig.1 (page 8 of the main manuscript).

3.5 Line 141: The author mention that it is possible to envisage many ways to construct atomic pair descriptor vectors. The authors clarify their chosen method, but I wonder if the author can provide any further comments on other options they tried whilst developing the model? This would be very useful to the community. I think this could be added to SI and hence not impact the text greatly in the main manuscript.

As mentioned by the reviewer, the adequate selection and construction of input features are of utmost importance in AI applications, having a remarkable impact on the actual performance of the resultant models. In the particular case of chemistry, chemical featurization (how to transform a molecule into a machine-readable array) is one of the major bottlenecks in developing chemical AI tools. We exploit the end-to-end nature of the SchNet representation to "learn" the atomic environment vectors on the fly, which can be fed directly into one-particle (1p) SchNet4AIM models. For two-particle quantities, we first opted for the straightforward concatenation of SchNet atomic features. However, doing so does not explicitly include direct information of a given atomic pair. The inter-atomic distance was embedded as an additional feature to overcome such issues, resulting in better-behaved and more accurate models. Such a finding is not unexpected as applying certain transformations on the input data is known to affect both the training times and accuracy, as happens with the Fourier features (see, for example, <https://doi.org/10.48550/arXiv.2006.10739>). Due to time constraints, we have not yet explored other alternative pairwise representation schemes. Nevertheless, this is a very relevant point which should be addressed in upcoming works.

Following the reviewer's suggestion, we have expanded our original discussion on the SchNet4AIM representation of atomic pairs and the construction of pairwise SchNet4AIM features. Changes are highlighted in blue on page 6 of the supplementary material (Supplementary Note 2).

3.6 Line 205: The author's generate several models during the course of the paper, but the only comparison in terms of the amount of data to train the model Vs. the models accuracy is a comparison some of the authors previous neural network model, as far as I can see. Could they comment upon the effect of training data set size on these models?

We agree with the reviewer, analyzing the evolution of the prediction accuracy with the size of the training dataset employed can provide very valuable insights into the learning abilities of the underlying models. As such, and following their suggestion, different SchNet4AIM models were trained to predict the atomic charges of the QTAIM electronic database using an increasing number of molecular instances, randomly selected from the original training pool. All other parameters, as well as the testing and validation subsets, were fixed for the sake of reliability and reproducibility. The results, collected in Supplementary Note 19, show that SchNet4AIM exhibits a fairly fast learning curve, reaching a plateau in the testing prediction accuracy for about 1700 molecular instances. Such a finding evidences the prominent generalization abilities of our models against never-seen data.

Following the reviewer's suggestion, we have included a discussion about the effect of the training dataset on the SchNet4AIM models. These results can be found in the newly added Supplementary Note 19 (in the supplementary material) as well as in the main text (lines 269-272, highlighted in blue).

3.7 The GitHub link seems to lead to a 404 gateway meaning we can not see the code or pre-trained models in the review <https://github.com/mgallegos/SchNet4AIM>.

We would like to thank the reviewer for checking the availability of the data and code supporting our findings. Although our code is open source, the GitHub repository hosting the data and code related to our work has been held private during the evaluation of this manuscript. However, the latter will now be made public. Having said that, we have included a compressed file with the contents of the GitHub repository so that it can be easily accessed by the reviewers.

3.8 It is not clear from the data availability statement whether the authors intend to make the training data available in full. I would urge the author to do so to enable the community to build on the work in addition to making the work highly transparent.

All the databases used in this work will be available to the reader. We hope that, as already addressed by the reviewer, this enhances the transparency of our work while actively contributing to the scientific community. Besides that, the SchNet4AIM repository also includes built-in examples for predicting QTAIM electron metrics of CHON neutral molecules (with uncertainty estimation), and a script to show the outputs of our models can be easily interpreted. The reviewer can find this at /SchNet4AIM/src/SchNet4AIM/examples/ in the GitHub repository, which is already public, (or /SchNet4AIM-main/src/SchNet4AIM/examples/ in the compressed file).

Besides this, the "Data Availability" statement has been updated accordingly for clarity, which is highlighted in blue on page 23 of the manuscript.

3.9 Typos and references: References 542 and 561 seem to be repeated. Line 249 “on.The” missing space

Following the reviewer’s suggestions and comments, we have addressed the typos in the main text and references. Changes can be found in blue, in lines 222 (page 11) and 297 (page 14).

Reviewer #4 (Remarks to the Author):

The authors propose and evaluate an interesting modification of SchNet with 1P and 2P descriptors. They demonstrate performance in global and local predictions tasks, extrapolation and interpretability tasks. The results are promising and the provided evaluation set ups are convincing. The paper is well-written and provides just the right amount of detail for a non QTAIM expert to understand the fundamental points. Also, I notice the SI is well documented and it looks like the final code implementation will be very useful. Overall, I recommend the work for publication and suggest the authors to address these considerations:

We would like to express our most sincere gratitude to the reviewer for their comments and evaluation of our work. In the upcoming section, details on how we have addressed their concerns are provided.

4. 1. The motivation and method for SchNet4AIM are clearly explained. However, it took me a careful review to identify 1) main modifications compared to ScNet and 2) explainability piece. I think the authors may benefit by modifying Figure 1 to highlight the main modifications of the ScNet architecture (1P and 2P descriptors, different outputs), as you are currently doing with the "pooling layer". Also, you could identify the "explainable" elements in the schematic.

Following the suggestion of the reviewer, we have updated Fig. 1 to provide a more detailed view of the main modifications made to the original SchNet work. Besides this, special emphasis has been made to highlight the explainability arising from the combination of real-space approaches and SchNet4AIM. Changes can be found in Fig.1 of the main manuscript (page 8).

4. 2. The model you are proposing is intrinsically explainable -- as the model outputs can be used for explainability. This is nicely demonstrated in the final case study. However, this is not clearly stated in the text. I suggest you mentioned this and comment you are not using an extrinsic interpretability or proving technique to provide explanations. A few sentences explaining how the model provides interpretable outputs and how to use these predictions, may be enumerating some potential use cases, can be useful for non-expert readers.

The reviewer is correct; in our approach, the explainability emerges naturally from using physically rigorous and consistent local quantum chemical properties, making the presented SchNet4AIM models intrinsically explainable and transferable. We have expanded our discussion on the interpretability of the SchNet4AIM outputs, which can be found in Supplementary Note 18 and the main text (in blue, in lines 132-134, 324-328, 422-423, and 442-454).

Moreover, we have included a script in the SchNet4AIM GitHub repository to ease the interpretation of the SchNet4AIM explainable outputs (more details are provided in point 4.10).

4. 3. The extrapolation task shows very promising results and I agree with the authors about the predictive power of 1P and 2P features. What are the limitations to this extrapolation capabilities?

We expect the extrapolation abilities of SchNet4AIM to be limited by three main factors:

-The size and quality of the training dataset: presenting the model with an extensive, varied collection of training instances can aid it in distilling the hierarchical representations that enable the mapping from the input features to the output properties. In this way, adequate sampling of the chemical space can result in more robust and reliable SchNet4AIM models with better generalization abilities, even in extrapolation regimes.

-The representation kernel employed: Relying on a local representation, SchNet4AIM has different parameters that allow control of the extent to which the local environment of an atom extends in space. Selecting a small to medium-sized cutoff radius will generally enhance the transferability of SchNet4AIM predictions, often at the expense of reducing the accuracy with which the local environment vectors describe the atomic and interatomic features.

-Finally, the extrapolation abilities are also highly dependent on the complexity of the target properties and the influence that the environment has on the latter.

That said, the extrapolation capabilities will be explored in further detail in upcoming works. Following the reviewer's suggestion, this topic has been explicitly discussed in the manuscript. Changes can be found in the supplementary material (Supplementary Note 13, in blue) and main text, lines 273-285.

The authors mentioned that larger and more complex systems will be more challenging, but is there any quantification or observation of these based on the modeled systems / case studies?

As mentioned by the reviewer, we generally expect larger and more complex systems to be more challenging. SchNet4AIM is more likely to encounter never-seen local chemical environments which are very distant to those found throughout the learning stage, and thus worsening the quality of the predictions. This can be readily seen in the main manuscript: for instance, despite the good consonance of Fig. 4, the prediction accuracy in these extrapolation domains is, as expected, lower than that found across the testing domains. This is directly evidenced in the performance metrics, shown in Supplementary Table 21. Likewise, and despite the great qualitative match between quantum chemically computed data and SchNet4AIM estimations, the prediction accuracy in the supramolecular systems is also below the training and testing model error metrics. This is directly reflected in Supplementary Note 16, which shows certain quantitative offsets in the SchNet4AIM predictions.

Addressing the reviewer's comment, we have explicitly discussed this topic. Changes can be found in the supplementary material (Supplementary Note 13, in blue) and the main text (278-281).

How do the authors balance / regularize the model complexity in such cases? I saw a brief mention of it in the SI, but I would appreciate clear discussion in the main manuscript.

Currently, different approaches are used to regularize the SchNet4AIM models, preventing them from excessively fitting the data and thus enhancing their performance in complex scenarios. First of all, we tune the SchNet4AIM representation parameters: using more compact

(lower number of features) and more short-sighted (reducing the cutoff radius or number of interaction layers) features enhances the generalization abilities of the models, even in spite of affecting the mean model performances. On top of that, a molecular-to-local trade-off is introduced in the loss function. The latter controls the strength with which SchNet4AIM is penalized when deviating from the ideal local values as well as from the reconstruction of the corresponding molecular observables. Increasing this trade-off prevents the model from excessively focusing on the local values, enhancing its ability to generalize to never-seen intricate systems. Finally, different loss function kernels, including the L1 and L2 norms, are used to obtain more flexible cost functions, effectively working as an elastic regularization kernel.

Following the reviewer's suggestion, this topic has been explicitly discussed in our work. Changes can be found in the supplementary material (Supplementary Note 13, in blue) as well as in the main text (lines 282-285).

4. 4. Also, in the extrapolation task, the authors used 10% of the dataset for training, is that enough to be defined as extrapolation? For me this is a few shot learning task, but I understand this can be defined different in the literature for the specific problem. What happens when training with less data?

In the “extrapolation task,” we train SchNet4AIM on 10% of an existing QTAIM electronic metric database comprising near-equilibrium CHON molecules. Then, we use the previously trained SchNet4AIM models to predict chemical reactions involving high-energy regions of the potential energy surface (PES). The latter are entirely outside the interpolation domains seen by our models during their training and thus comprise a clear case of extrapolation. To conclude, the extrapolation feature does not come from using 10% of the database but from testing our model in never-seen regions of the PES, well beyond the interpolation domains of the underlying models.

We introduced changes clarifying this issue to the manuscript. Changes can be found in blue on page 11 (lines 228-230) of the main manuscript.

4. 5. What are the model limitations for further use in the SchNet applications?

In the scenario of real-space techniques, the framework where SchNet4AIM has been derived, there are two significant limitations related to the reference data. On the one hand, computing the reference data entails a substantial computational cost. This hampers both the creation of the training dataset and the evaluation of the SchNet4AIM model performance in large scenarios (where the quantum-chemically computed values may not even be accessible in feasible timescales). On the other hand, the conventional calculation of some real-space terms often involves the numerical evaluation of a very large number of multi-dimensional integrals. Doing so can easily result in non-negligible integration errors, which become especially notorious for large and complex systems out of equilibrium. Hence, having access to high-quality reference data is the major bottleneck in developing SchNet4AIM models that predict real-space quantities.

We have included a brief discussion on the SchNet4AIM model limitation in the main text, which can be found, highlighted in blue, in lines 457-459.

4. 6. Does the training datasets require any specific features compared to traditional datasets used to train SchNet in multiple applications?

SchNet4AIM exclusively requires molecular geometry, given in terms of XYZ Cartesian coordinates, as the only input feature to be provided by the user (see Supplementary Notes 2 and 3 or lines 155-157 of the main text). Being an end-to-end model, SchNet4AIM transforms the latter on the fly into a more suitable hierarchical representation, relying on the so-called SchNet features. Furthermore, the inter-atomic distances are explicitly included as an additional feature when dealing with two-particle target properties. All of this is directly handled and computed inside the model. As for the actual formatting of the database, SchNet4AIM can deal with both “db” and “JSON” formats. This has been specified in the Methods section of the main text, lines 489-491, in blue.

4. 7. Is permutation invariance in 2P features enough or could the model be further expanded with translational or rotational invariance? I had to look at the original SchNet paper to understand this piece so it might be good clarify this.

All the constituents of our 2P features (the inter-atomic distance and the SchNet atomic representation) are permutationally, translationally, and rotationally invariant. In this way, both the 1P and 2P features used by SchNet4AIM fulfill all three (permutational, translational, and rotational) invariances. For the sake of clarity and readability, corresponding modifications have been added to the manuscript. Changes can be found in blue in the main text (page 8, lines 163-164).

4. 8. What are potential model improvements?

The models presented here are just proof of concept to show the potential and applicability of SchNet4AIM in the interpretable estimation of rigorous local quantum chemical properties, along with their reconstructed molecular analogs. As such, there is still much room for improvement, especially in model hyper-parameters and training procedures. For instance, fine-tuning the cutoff radius used in the SchNet4AIM representation is expected to be crucial in controlling the compromise between prediction accuracy and transferability. Implementing active learning protocols is particularly useful to improve the extrapolation abilities of the underlying models. However, these are outside the scope of this work and should be addressed independently.

A clarifying sentence regarding potential model improvements has been included in the main text. The latter can be found in blue, in lines 457-461.

4. 9. Regarding explanations, I think the interpretability case study is great illustration of the mode capabilities. What are some limitations in terms of predictive power and explainability? Can those be discussed more broadly in the paper along with potential explainability use cases?

Following the reviewer's suggestion, we have expanded our discussion on the explainability of our model along with its potential use cases. Changes can be found in blue, in lines 442-454 of the main text. We want to stress that a comprehensive answer to this question would require many practical applications, which need time and the model's release in the first place.

4. 10. Model explanations as outputs are very useful, but are there limitations to them? For example, defining more specific / macro interpretability physical quantities that are not as clear connected to atomistic interaction? Can the features / learned representations be interpreted too? Are extrinsic explainability techniques recommended or necessary?

We thank the reviewer for bringing up this fascinating topic. Although interpretable ML outputs are of great importance and utility, they have certain limitations. It is worth recalling at this point that this is something inherent to XCAI, as this field is still in its infancy.

In the particular case of the SchNet4AIM models presented so far, one can identify two main challenges in their explainability. Firstly, the number of atomic pairs grows rapidly as $(N*(N-1)/2)$, being N the number of atoms) with the size of the system. As such, identifying the most relevant pairwise interactions dominating a given process becomes more difficult with the supramolecular system's complexity, especially when studying the interaction between extensive molecular moieties. Secondly, our XCAI implementation is currently limited to explaining how their local components dominate molecular or group quantities. However, SchNet4AIM, with other techniques such as extrinsic explainable approaches, could shed light on this last point.

As for the macro interpretability, our SchNet4AIM models are particularly privileged since the underlying QTAI underpinnings could eventually allow us to map the overall behavior of a molecular property to the intricate interaction between different sub-groups and moieties in the system.

Furthermore, we have included a script to ease the interpretation of the SchNet4AIM outputs, exemplified by identifying the pairwise terms that dominate the behavior of the group electron delocalization between two molecules or chemical moieties across a dynamic scenario. This can be found in our GitHub repository (or in the compressed file at /SchNet4AIM-main/src/SchNet4AIM/examples/scripts/explain_group_DI.py).

Following the reviewer's suggestion, our work thoroughly discusses this topic. Changes can be found, highlighted in blue, at the end of Supplementary Note 18 and in the main text (lines 447-454).

REVIEWERS' COMMENTS

Reviewer #1 (Remarks to the Author):

I am pleased that the authors have addressed all my comments to my satisfaction. They have substantially revised and expanded their original manuscript. I recommend that their manuscript be published.

Reviewer #3 (Remarks to the Author):

The author's have addressed my comments very well. I would like to thank them for taking the time to address these comments and in particular for introducing the uncertainty metrics, which I think is a valuable addition.

I believe the manuscript should be published.

Reviewer #3 (Remarks on code availability):

I have looked over the code on GitHub. The code is generally well documented with an extensive README and permissive license. The code could be improved with the use of type hinting. I was unable to run the code due to the lack a cuda library suitable for my local machine (Mac M1).

Reviewer #4 (Remarks to the Author):

The author addressed all my points.

I specially appreciate the expanded discussion and the additional script for interpretability in the GitHub repo. I recommend to go ahead with publication.

Reviewer #4 (Remarks on code availability):

Yes, very detailed instructions and examples.